# Lymph node stromal cells constrain immunity via MHC class II self-antigen presentation

**Antonio P Baptista[1,2†], Ramon Roozendaal[1†‡], Rogier M Reijmers[1], Jasper J Koning[1], Wendy W Unger[1], Mascha Greuter[1], Eelco D Keuning[1], Rosalie Molenaar[1], Gera Goverse[1], Marlous M S Sneeboer[1], Joke M M den Haan[1], Marianne Boes[3], Reina E Mebius[1]\***

[1]Department of Molecular Cell Biology and Immunology, Vrije Universiteit Medical Center, Amsterdam, Netherlands; [2]Graduate Program in Areas of Basic and Applied Biology, University of Porto, Porto, Portugal; [3]Department of Pediatric Immunology, Laboratory of Translational Immunology, University Medical Center Utrecht, Utrecht, Netherlands

**\*For correspondence:** r.mebius@vumc.nl

†These authors contributed equally to this work

**Present address:** ‡Infectious Diseases and Vaccines Therapeutic Area, Janssen Pharmaceutical Companies of Johnson and Johnson, Leiden, Netherlands

**Competing interests:** The authors declare that no competing interests exist.

**Reviewing editor**: Emil R Unanue, Washington University School of Medicine, United States

**Abstract** Non-hematopoietic lymph node stromal cells shape immunity by inducing MHC-I-dependent deletion of self-reactive CD8+ T cells and MHC-II-dependent anergy of CD4+ T cells. In this study, we show that MHC-II expression on lymph node stromal cells is additionally required for homeostatic maintenance of regulatory T cells (Tregs) and maintenance of immune quiescence. In the absence of MHC-II expression in lymph node transplants, i.e. on lymph node stromal cells, CD4+ as well as CD8+ T cells became activated, ultimately resulting in transplant rejection. MHC-II self-antigen presentation by lymph node stromal cells allowed the non-proliferative maintenance of antigen-specific Tregs and constrained antigen-specific immunity. Altogether, our results reveal a novel mechanism by which lymph node stromal cells regulate peripheral immunity.

## Introduction

Lymph nodes play a crucial role in the initiation of adaptive immune responses by bringing together antigens, antigen-presenting cells (APCs), and antigen-specific lymphocytes (*Junt et al., 2008*; *Mueller and Germain, 2009*; *Roozendaal and Mebius, 2011*). The interaction between these components within the lymph node parenchyma is fostered by its highly structured architecture, which is dictated by resident non-hematopoietic stromal cells (*Katakai et al., 2004*). Despite their low frequency among total lymph node cells, lymph node stromal cells have gathered considerable attention recently, as they were found to have important immunoregulatory functions. Indeed, in addition to promoting the communication between dendritic cells (DCs) and lymphocytes by providing anchorage for DCs and guiding T cells via the production of chemokines (*Sixt et al., 2005*; *Bajenoff et al., 2006*), lymph node stromal cells were shown to regulate the size of the lymphocyte pool by providing essential survival factors (*Link et al., 2007*) and to limit T cell activation/proliferation by producing nitric oxide (*Lukacs-Kornek et al., 2011*; *Siegert et al., 2011*).

In addition to these broad mechanisms that control the entire lymphocyte pool, lymph node stromal cells were also found to influence lymphocytes in a cognate antigen-dependent manner. Similar as to thymic epithelial cells, several subsets of lymph node stromal cells were shown to express and present peripheral tissue-restricted antigens (PTAs) (*Nichols et al., 2007*; *Cohen et al., 2010*; *Fletcher et al., 2010*). Presentation of PTA-derived peptides in the context of MHC-I molecules leads to the deletion of auto-reactive CD8+ T cells (*Lee et al., 2007*; *Nichols et al., 2007*; *Magnusson et al., 2008*;

**eLife digest** In vertebrates, the immune response that protects against infection and disease is made up of two systems. The body's first line of defense is the innate immune system that attacks invaders rapidly but indiscriminately. If this fails to stop disease progression, the adaptive immune system is activated. Although the adaptive immune response is relatively slow compared with the innate immune response, it is more deliberate and produces cells that specifically target and destroy the pathogen or diseased cells present. The adaptive immune system also produces cells that 'remember' the pathogen so that it can be destroyed more quickly if it invades again.

A special type of white blood cell, called a T cell, is key to the adaptive immune response. To activate T cells, fragments of molecules that provoke an immune response—called antigens—must be bound to a 'major histocompatibility complex' (MHC) and presented to these cells. This process often occurs in lymph nodes, organs that filter the fluid moving from the body's tissues back into the blood.

Particular cells in the lymph node, called lymph node stromal cells, are essential for the organ's structure; recently, these cells have also been found to play roles in regulating the immune response. For example, lymph node stromal cells can help to destroy self-reactive T cells that attack the host's normal, healthy cells. In addition, some types of lymph node stromal cells produce major histocompatibility complexes, although exactly what these complexes do on these cells was unknown.

Baptista, Roozendaal et al. investigated the role of the major histocompatibility complexes expressed by lymph node stromal cells by transplanting mutant cells that could not produce these complexes into otherwise normal mice. In these mice, T cells became more activated than normal and the transplant was rejected after several weeks.

On further investigation, Baptista, Roozendaal et al. discovered that the major histocompatibility complexes produced by the lymph node stromal cells help to maintain an active population of regulatory T cells. These cells are responsible for shutting down the immune response. This work therefore improves our understanding of how the immune response is regulated and could help to develop new strategies for preventing donor organs being rejected after transplantation.

*Yip et al., 2009*; *Cohen et al., 2010*; *Fletcher et al., 2010*). Furthermore, acquisition of dendritic cell-derived peptide–MHC-II complexes by lymph node stromal cells gives them the ability to negatively regulate CD4[+] T cell proliferation and survival (*Dubrot et al., 2014*).

Importantly, next to the ability to capture MHC-II molecules, lymph node stromal cells were also shown to express their own MHC-II molecules (*Malhotra et al., 2012*; *Dubrot et al., 2014*). In this study, we have investigated, for the first time, the function of endogenous MHC-II molecules on lymph node stromal cells. We found MHC-II expression on lymph node stromal cells to be instrumental for the maintenance of FoxP3[+] regulatory T cells (Tregs), thereby safeguarding the homeostasis of the immune system by limiting immune reactivity. Altogether, our data add a new layer to the immunoregulatory properties of lymph node stromal cells.

## Results

### Lymph node stromal cells express surface MHC-II

MHC-II expression was traditionally thought to be restricted to hematopoietic-derived professional antigen-presenting cells, such as dendritic cells, macrophages, and B cells. It is now clear, however, that cells of other origins are also able to express MHC-II molecules and to present antigens to CD4[+] T cells (*Stagg et al., 2006*; *Kreisel et al., 2010*; *Koyama et al., 2012*). Amongst these other cells, lymph node stromal cells express MHC-II in the steady-state (*Figure 1* and (*Malhotra et al., 2012*; *Dubrot et al., 2014*)). By flow cytometry, we found CD45[−]gp38[+]CD31[−] fibroblastic reticular cells (FRCs), CD45[−]gp38[+]CD31[+] lymphatic endothelial cells (LECs), and CD45[−]gp38[−]CD31[+] blood endothelial cells (BECs) to express MHC-II on their surface (*Figure 1A*). In agreement, these cells also contained mRNA transcripts for MHC-II (H2-Ab1) itself as well as for the MHC-II-related molecules CD74 (invariant chain—Ii), H2-M (chaperone that catalyzes peptide loading onto MHC-II molecules), and LAMP-1 (marker for endosomes/lysosomes, where antigen processing and MHC-II loading occur)

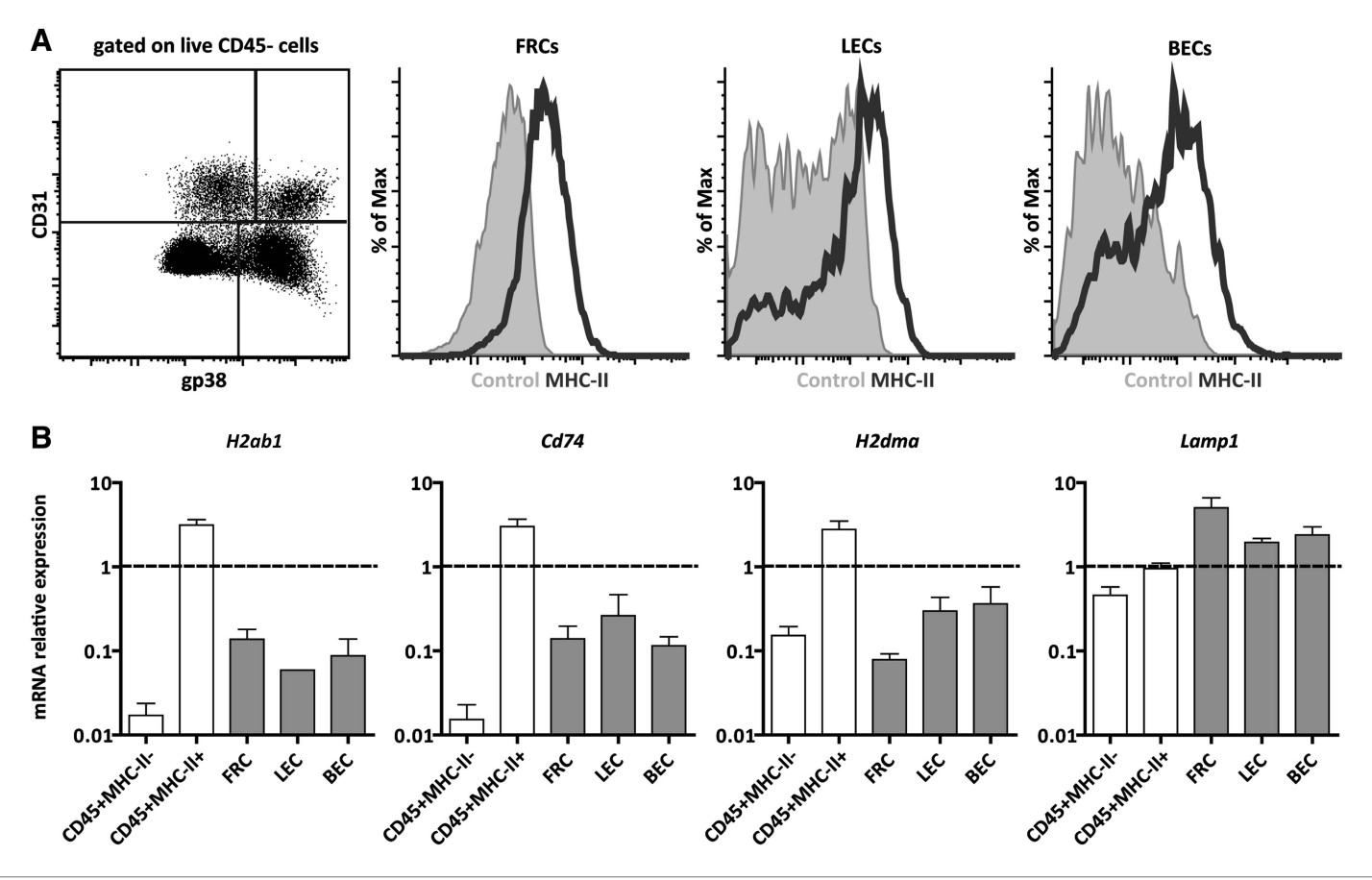

**Figure 1**. Lymph node stromal cells express MHC-II in the steady-state. (**A**) MHC-II expression on lymph node stromal cells was assessed by flow cytometry. Fibroblastic reticular cells (FRCs) were identified as CD45⁻gp38⁺CD31⁻ cells; lymphatic endothelial cells (LECs) as CD45⁻gp38⁺CD31⁺ cells; and blood endothelial cells (BECs) as CD45⁻gp38⁻CD31⁺ cells. Filled histograms represent control staining, whereas open histograms represent MHC-II expression. Representative example of five independent experiments performed. (**B**) mRNA expression of MHC-II (H2-Ab1) and MHC-II-related genes, CD74, H2-M, and LAMP-1, was determined on wild-type FACS-sorted stromal cells by real-time PCR. Total pLN cells (arbitrarily set at one) and FACS-sorted CD45⁺MHC-II⁻ and CD45⁺MHC-II⁺ cells were used as controls. The data represent mean ± SEM; n = 4.

(*Figure 1B*). Altogether, these data suggest that lymph node stromal cells possess the necessary machinery to process and present antigens in the context of MHC-II molecules.

## MHC-II expression on lymph node stromal cells regulates T cell activation

To determine the function of MHC-II expression on lymph node stromal cells, we performed lymph node transplantation experiments in which popliteal lymph nodes of wild-type recipient mice were surgically replaced by MCH-II knock-out (KO) lymph nodes. Within 4 weeks, these transplants reconnect to the lymphatic and blood vasculature. While the stromal cell compartment within the transplant remains of donor origin, virtually all immune cells of the donor animal will be replaced by host-derived cells (*Wolvers et al., 1999*; *Hammerschmidt et al., 2008*; *Molenaar et al., 2009*). Therefore, transplantation of MHC-II KO lymph nodes into wild-type recipients resulted in selective absence of stromal cell endogenously expressed MHC-II molecules, whereas MHC-II was normally expressed on the recipient-derived hematopoietic cells that had migrated into the transplanted lymph node (*Figure. 2—figure supplement 1*). Residual MHC-II expression on stromal cells (*Figure. 2—figure supplement 1*) was likely due to the acquisition of dendritic cell-derived peptide–MHC-II complexes (*Dubrot et al., 2014*). Of note, extra-thymic AIRE-expressing cells (ETACs), which were initially characterized as a stromal cell subset (*Gardner et al., 2008*) but later determined to be hematopoietic-derived (*Gardner et al., 2013*),

expressed CD45, CD11c, EpCAM, as well as MHC-II (*Figure 2—figure supplement 2*) within MHC-II KO lymph node transplants (*Figure 2—figure supplements 1,2*). Therefore, ETACs are likely not involved in the phenotypes described below. Ablation of MHC-II on lymph node stromal cells resulted in increased frequencies of CD62L⁻CD44⁺ activated CD4⁺ and CD8⁺ T cells within the transplanted lymph nodes (*Figure 2A*). T cell activation was restricted to MHC-II KO transplants, as the endogenous lymph nodes of MHC KO lymph node transplant recipients showed similar frequencies of naïve and activated T cells as compared to the endogenous lymph nodes of wild-type transplant recipients (*Figure 2—figure supplement 3*). Likely as a consequence of local activation, MHC-II KO transplants ended up being rejected, since by week 8 post-transplantation, they could not be recovered. Compared to wild-type transplants, 4 weeks after transplantation, MHC-II KO lymph node transplants showed significant rigidity and tissue damage and the unusual presence of large clusters of CSFR1⁺CD11b⁺ CD11c⁺Moma2⁺F4/80⁺MHC-II⁺ macrophages (*Figure 2—figure supplement 4*). Furthermore, they showed increased expression of several gene transcripts recently associated with a 'universal' rejection module (*Khatri et al., 2013*) and an overall increased common rejection module (CRM) score (*Figure 2—figure supplement 5*). Altogether, our results suggest that MHC-II expression on lymph node stromal cells regulates T cell activation, exerting a local dampening effect on both CD4⁺ and CD8⁺ T cells.

## CD4⁺ T cells restrict CD8⁺ T cell activation in MHC-II KO lymph node transplants

As MHC-II molecules are not thought to directly mediate cellular interactions with CD8⁺ T cells, we reasoned that CD8⁺ T cell activation in the absence of lymph node stromal cell MHC-II expression could be an indirect effect of local CD4⁺ T cell activation. We tested this hypothesis by depleting CD4⁺ cells with bi-weekly intraperitoneal injections of the anti-CD4 antibody GK1.5, starting 1 week before transplantation until the time of analysis (4 weeks after transplantation) (*Figure 2—figure supplement 6*). In contrast to our expectation, CD4⁺ T cell depletion led to a further increase in the frequency of activated CD62L⁻CD44⁺ CD8⁺ T cells in MHC-II KO transplants (*Figure 2B*). These results therefore suggested that CD8⁺ T cell activation in the absence of MHC-II expressing lymph node stromal cells was not a direct consequence of deregulated CD4⁺ T cell activation, but rather appeared to be constrained by CD4⁺ T cells. Further supporting this notion, also within the endogenous lymph nodes of CD4-depleted mice receiving MHC-II KO lymph node transplants, a significant increase of CD62L⁻CD44⁺ CD8⁺ T cells was observed when compared to wild-type lymph node transplant recipients (*Figure 2—figure supplement 7*). Thus, it appears that in contrast to our initial hypothesis, CD4⁺ T cells not only restrain local CD8⁺ T cell activation in transplanted lymph nodes in a manner that is dependent on lymph node stromal cell endogenous MHC-II expression but are also required to prevent its systemic spreading.

## MHC-II⁺ stromal cells support FoxP3⁺ Treg proliferation

T cell activation is largely kept in check by Treg cells, thereby safeguarding the homeostasis of the immune system. Since Treg frequency was reduced in MHC-II KO lymph node transplants (*Figure 2—figure supplement 6*) and Treg development and maintenance involves agonistic selection on MHC-II presented peptides (*Josefowicz et al., 2012*), we decided to assess directly whether lymph node stromal cell endogenous MHC-II deficiency would affect Tregs. By transferring CFSE-labeled T cells into *Rag2⁻/⁻* mice transplanted with either MHC-II KO or wild-type lymph nodes, we found the homeostatic proliferation of CD4⁺FoxP3⁺ cells to critically depend on stromal cell endogenous MHC-II expression. As compared to wild-type transplants, CD4⁺FoxP3⁺ cells proliferated roughly 3 times less efficiently in MHC-II KO lymph node transplants (*Figure 3A,B* and *Figure 3—figure supplement 1*), resulting in a threefold reduction in Treg frequency (*Figure 3C*). A reduction in the proliferation of CD4⁺FoxP3⁻ cells was also observed in MHC-II KO lymph node transplants, although this did not reach statistical significance (*Figure 3B* and *Figure 3—figure supplement 1*). Overall, these results revealed a pivotal role for lymph node stromal cell MHC-II expression on the homeostatic proliferation of CD4⁺ T cells, which seemed particularly relevant for the homeostatic maintenance of CD4⁺Foxp3⁺ Tregs.

## Presentation of endogenous self-antigens by lymph node stromal cells supports Treg maintenance in vitro

The observation that MHC-II expression on lymph node stromal cells impacted on the peripheral maintenance of Tregs implied that cognate ligands for the T cell receptor (TCR) of Tregs are expressed and

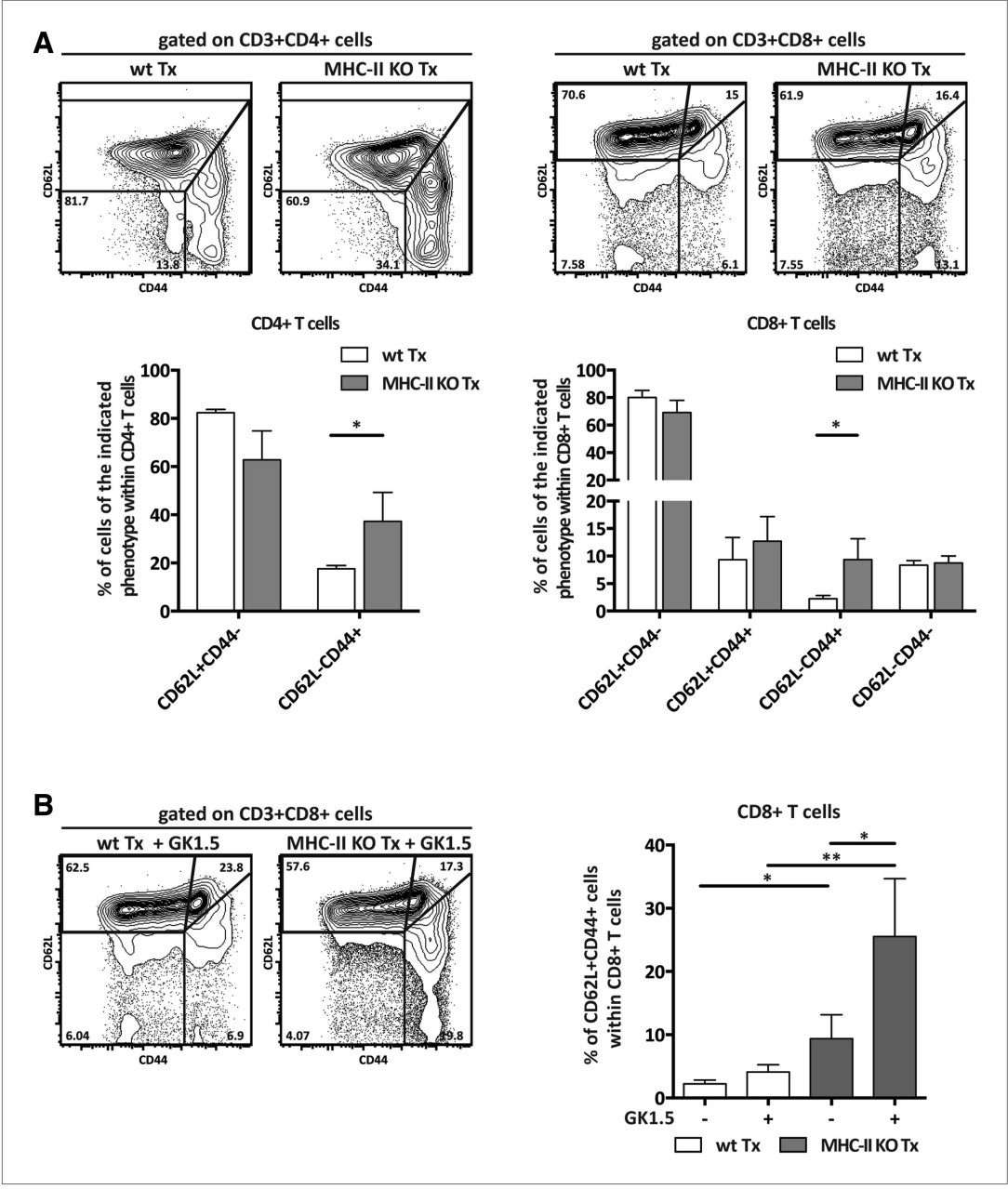

**Figure 2**. MHC-II+ lymph node stromal cells regulate T cell activation. Wild-type mice were transplanted with either wild-type (wt Tx) or MHC-II KO (MHC-II KO Tx) lymph nodes. After 4 weeks, host-derived CD4+ and CD8+ T cells present within the transplants were characterized by flow cytometry (**A**). In (**B**), transplanted animals were depleted of CD4+ T cells by administration of the antibody GK1.5. For easier comparison, the data of ***Figure 2A*** regarding CD62L−CD44+CD8+ T cells are duplicated here. Representative contour plots are shown; the numbers in the plots indicate the frequency of cells within the drawn gates. The data represent mean ± SEM; n = 4; *p ≤ 0.05, **p ≤ 0.01.

The following figure supplements are available for figure 2:

**Figure supplement 1**. Stromal cell MHC-II expression in MHC-II KO lymph node transplants.

**Figure supplement 2**. Lymph node EPCAM+ cells phenotypically represent a dendritic cell subset that is still present in MHC-II KO lymph node transplants as MHC-II expressing cells.

*Figure 2. Continued on next page*

*Figure 2. Continued*

**Figure supplement 3**. In normopenic conditions, T cell activation is restricted to MHC-II KO lymph node transplants.

**Figure supplement 4**. MHC-II KO lymph node transplants are rejected in wild-type recipients.

**Figure supplement 5**. MHC-II KO lymph node transplants exhibit increased common rejection module (CRM) score.

**Figure supplement 6**. Efficient CD4+ T cell depletion in donor and recipient tissues.

**Figure supplement 7**. CD4+ T cells prevent the systemic spreading of MHC-II-deficient stromal cell-mediated CD8+ T cell activation.

presented by the lymph node stromal cell compartment. In support of such hypothesis, previous research has reported expression of several peripheral tissue-restricted antigens (PTAs) in lymph node stromal cells (*Nichols et al., 2007*; *Cohen et al., 2010*; *Fletcher et al., 2010*). To assess whether lymph node stromal cells were able to present endogenous antigens in the context of MHC-II molecules, we used K14-mOVA transgenic mice, in which ovalbumin (OVA) expression is driven by the human keratin 14 promoter (*Bianchi et al., 2009*). In these mice, OVA is expressed in the skin and thymus (*Bianchi et al., 2009*) as well as in the 3 major lymph node stromal cell subsets (*Figure 4A*). As all primary lymph node stromal cell cultures established contained large amounts of contaminating hematopoietic cells (data not shown), which precluded our in vitro antigen presentation assays, we generated distinct lymph node stromal cell lines by long-term in vitro culture of lymph node single cell suspensions of K14-mOVA mice on collagen matrixes. Using this approach, we generated one cell line (K14-mOVAneg), resembling FRCs, which did not express detectable OVA mRNA transcripts and was thus used as a control cell line, and another cell line (K14-mOVApos), resembling LECs, which expressed OVA transcripts abundantly (*Figure 4B* and *Figure 4—figure supplement 1*). Co-culture of these cell

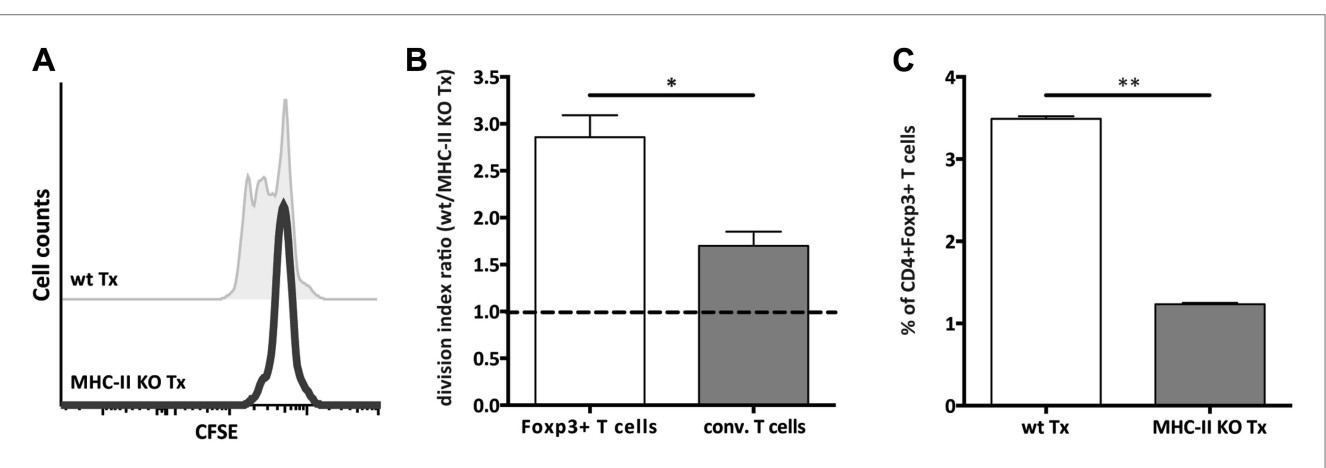

**Figure 3**. MHC-II+ lymph node stromal cells support Treg homeostatic proliferation. *Rag2*-deficient mice transplanted with either wild-type (wt Tx) or MHC-II KO (MHC-II KO Tx) lymph nodes were injected with $10^7$ CFSE-labeled wild-type lymphocytes. 48 hr later, mice were sacrificed and the transferred cells within the lymph node transplants analyzed by flow cytometry. The CFSE profile of transferred Foxp3+CD4+ T cells is shown in (**A**). In (**B**), the ratio between the division indexes of wild-type (B6) and MHC-II KO lymph node transplant recovered CD4+Foxp3+ Tregs and CD4+Foxp3− conventional T cells is shown. The frequency of CD4+Foxp3+ Tregs recovered from wild-type and MHC-II KO lymph node transplant is shown in (**C**). The data represent mean ± SEM; n = 2 independent experiments with 2–3 animals per group; *p ≤ 0.05, **p ≤ 0.01.

The following figure supplement is available for figure 3:

**Figure supplement 1**. Reduced CD4+ T cell proliferation in the absence of lymph node stromal cell MHC-II expression.

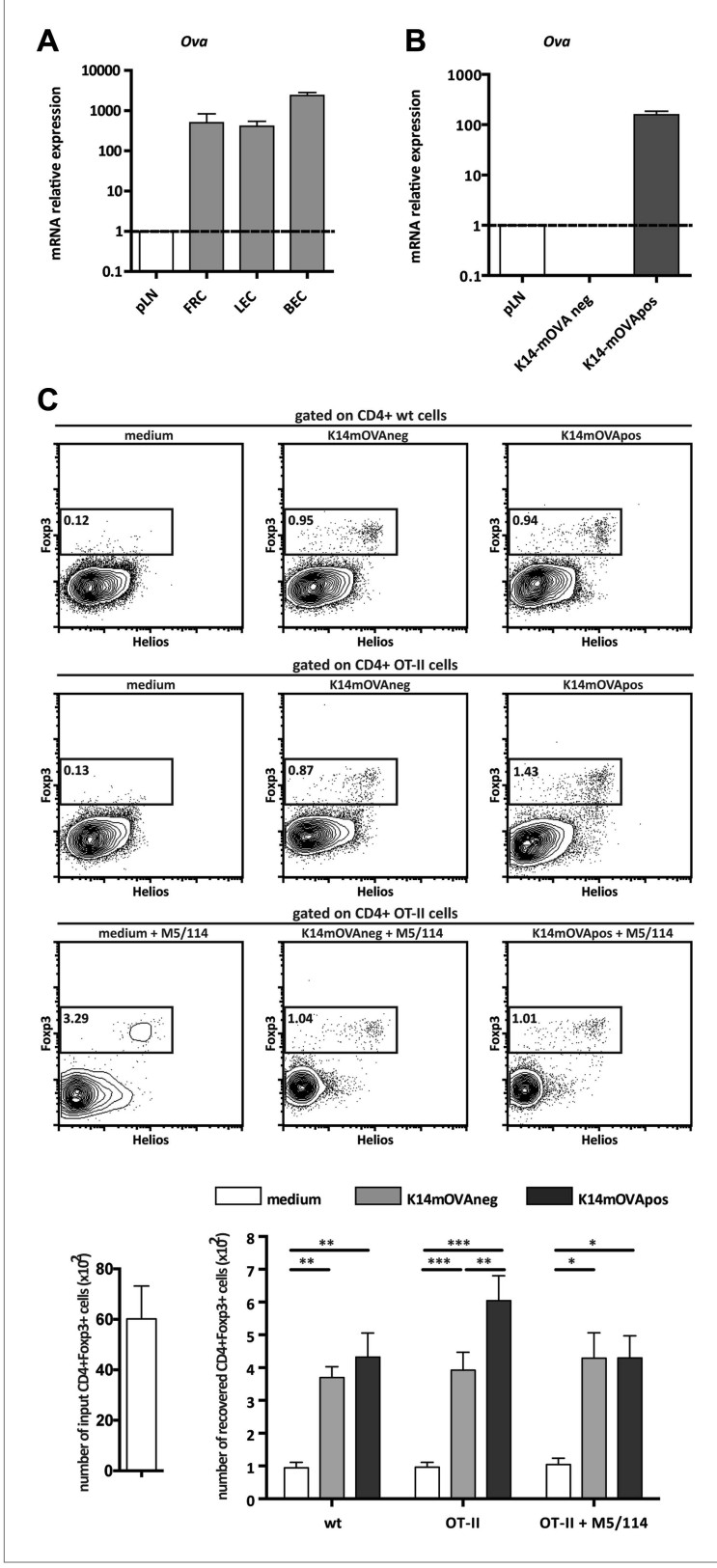

**Figure 4**. Endogenous OVA presentation by lymph node stromal cells promotes Treg maintenance in vitro. OVA mRNA expression in primary FACS-sorted stromal cells (**A**; n = 2) and in vitro-generated stromal cell lines derived from K14-mOVA mice (**B**; n = 5) was determined by real-time PCR. Peripheral lymph nodes (pLN) from
*Figure 4. Continued on next page*

*Figure 4. Continued*

K14-mOVA mice were used as controls. (**C**) MACS-sorted CD4+ wild-type or OT-II transgenic cells were cultured together with in vitro-generated stromal cell lines of K14-mOVA origin in the absence or presence of the MHC-II blocking antibody M5/114. After 72 hr of co-culture, OT-II cells were characterized by flow cytometry. Representative counterplots are shown; the numbers in the plots represent the frequency of CD4+Fox3+ T cells. Graphs depict the number of CD4+Fox3+ T cells in the beginning and at the end of culture. Data represent mean ± SEM; n = 3 for wild-type CD4+ T cells; n = 8 for OT-II cells; and n = 3 for OT-II cells + M5/114. *p ≤ 0.05, **p ≤ 0.01, ***p ≤ 0.001.

The following figure supplements are available for figure 4:

**Figure supplement 1**. Phenotypically K14-mOVAneg and K14-mOVApos cells represent fibroblastic reticular cells (FRCs) and lymphatic endothelial cells (LECs), respectively.

**Figure supplement 2**. K14-mOVA stromal cells present endogenous OVA-derived peptides in vitro.

**Figure supplement 3**. Endogenous OVA presentation by lymph node stromal cells does not affect CD4+Foxp3− conventional T cells.

lines with OVA-specific CD8+ OT-I and CD4+ OT-II T cells revealed that OVA-derived peptides could be presented by lymph node stromal cells in MHC-I as well as MHC-II molecules, as both OT-I and OT-II T cells showed increased CD25 expression when cultured with K14-mOVApos but not with K14-mOVAneg cells (*Figure 4—figure supplement 2*). Neither lymph node stromal cell line induced OT-I or OT-II T cell proliferation, however (*Figure 4—figure supplement 2*). To directly address whether self-antigen presentation by lymph node stromal cells in the context of MHC-II molecules influenced Tregs, we repeated our co-culture experiments and stained CD4+ T cells for Foxp3 and Helios. As compared to culture in medium alone, co-culture of CD4+ T cells with either cell line increased the survival of Foxp3+ cells irrespective of their TCR specificity (*Figure 4C*), which is suggestive of the production of T cell survival factors by both cell lines (*Link et al., 2007*). More importantly, K14-mOVApos cells significantly increased the recovery of Foxp3+ OT-II T cells as compared to the K14-mOVAneg cell line (*Figure 4C*). This effect was not apparent in co-cultures with wild-type CD4+ T cells and could be blocked by the addition of the MHC-II blocking antibody M5/114 (*Figure 4C*), indicating that the presentation of OVA-derived peptides in the context of MHC-II molecules by the K14-mOVApos cell line was the driver of increased Foxp3+ OT-II T cell survival in our assays. Of significance, CD4+Foxp3− conventional OT-II T cells, in the exact same conditions, did not show similar behavior (*Figure 4—figure supplement 3*). Overall, our data suggest that MHC-II-mediated self-antigen presentation by lymph node stromal cells drives CD4+Foxp3+ Treg maintenance.

## Presentation of endogenous self-antigens by lymph node stromal cells supports peripheral maintenance of Tregs in vivo

To address the effect of self-antigen presentation by lymph node stromal cells on antigen-specific Tregs in vivo, we transplanted lymph nodes of K14-mOVA mice into wild-type hosts and analyzed the fate of transferred OVA-specific OT-II CD4+ T cells. In this transplantation setting, OVA expression became confined to the transplanted lymph node stromal cells. Transplantation of K14-mOVA lymph nodes did not affect the endogenous pools of CD8+ T cells, CD4+ T cells, or CD4+FoxP3+ Tregs, as in the transplants as well as in the endogenous lymph nodes of the different 'chimeric' mice, the frequencies of the various T cell subsets were comparable (*Figure 5—figure supplement 1*). Similarly, transferred OT-II T cells showed a comparable distribution of naïve CD62L+CD44− and activated CD62L−CD44+ phenotypes in wild-type and K14-mOVA lymph node transplants as well as in wild-type endogenous lymph nodes (*Figure 5A,B*). Confirming our in vitro data, OT-II T cells did not proliferate (data not shown). Presentation of OVA by the lymph node stroma, however, led to a significant increase in the number of CD4+FoxP3+ OT-II T cells, particularly of the naïve-like CD62L+ subtype, within K14-mOVA transplants as compared to wild-type lymph node transplants (*Figure 5C*). Higher frequencies of total and CD62L+ naïve-like CD4+FoxP3+ OT-II Tregs were also evident in the endogenous lymph nodes of recipients of K14-mOVA lymph node transplants as compared to wild-type lymph node transplant recipient mice, although this did not translate in increased cell numbers (*Figure 5D*). In

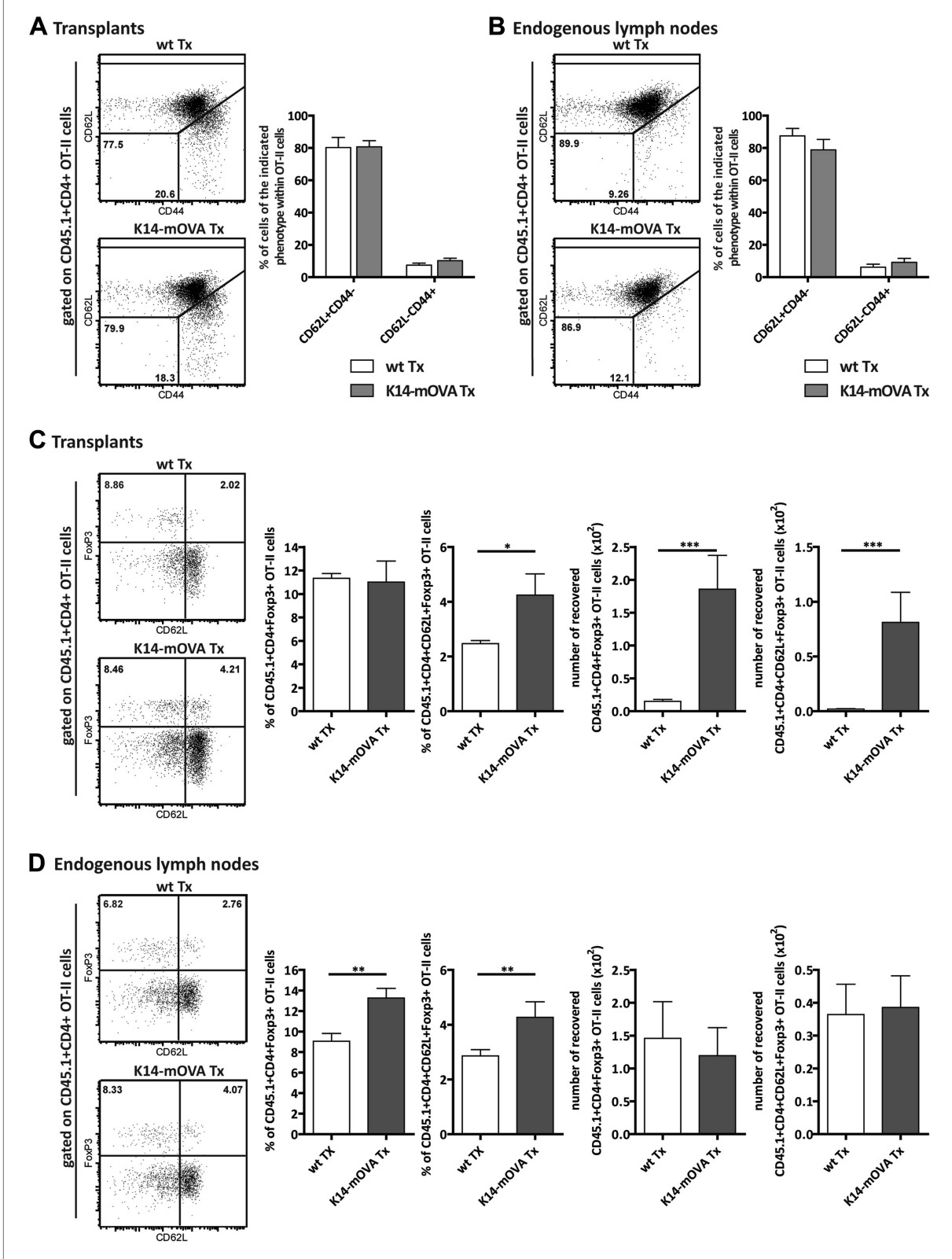

**Figure 5**. Endogenous OVA presentation by lymph node stromal cells promotes Treg maintenance in vivo. Wild-type mice transplanted with either wild-type (wt Tx) or K14-mOVA transgenic (K14-mOVA Tx) lymph nodes were injected with $10^7$ CD45.1+ OT-II cells. 3 days after the transfer, mice were sacrificed and the transferred OT-II cells in both the transplanted lymph nodes (**A**, **C**) and the endogenous lymph nodes (**B**, **D**) analyzed by flow

*Figure 5. Continued on next page*

*Figure 5. Continued*

cytometry. Naïve and activated OT-II T cells were defined as CD62L⁺CD44⁻ and CD62L⁻CD44⁺ cells, respectively (**A**, **B**); naïve-like OT-II Tregs as CD4⁺CD62L⁺Foxp3⁺ cells (**C**, **D**). Representative contour plots of the analysis performed are shown on the left; the numbers in the plots indicate the frequency of cells within the drawn gates. The graphs shown on the right represent the mean ± SEM of 2 independent experiments; n = 6 and n = 8 for wild-type and K14-mOVA lymph node transplanted animals, respectively; *p ≤ 0.05, **p ≤ 0.01, ***p ≤ 0.001.

The following figure supplements are available for figure 5:

**Figure supplement 1**. Endogenous CD8+ and CD4+ T cells are not affected by the transplantation of K14-mOVA transgenic lymph nodes.

**Figure supplement 2**. Expansion of CD4⁺FoxP3⁺Helios⁺ OT-II Tregs in mice transplanted with K14-mOVA lymph nodes.

conclusion, our data suggest that the presentation of endogenous antigens by the lymph node stroma contributes to the selective maintenance of antigen-specific Tregs.

## Presentation of endogenous self-antigens by lymph node stromal cells constrains immune reactivity

The increased frequencies and numbers of CD4⁺FoxP3⁺ OT-II T cells in K14-mOVA lymph node transplant recipients suggested that the presentation of OVA as a self-antigen by lymph node stromal cells may promote the development of tolerance towards OVA. To test this hypothesis, we measured OVA-specific delayed-type hypersensitivity (DTH) responses in mice transplanted with either K14-mOVA or wild-type lymph nodes. As determined by ear swelling, DTH responses towards OVA were significantly reduced in mice transplanted with K14-mOVA lymph nodes as compared to mice transplanted with wild-type lymph nodes (*Figure 6A*), suggesting that OVA presentation by lymph node stromal cells restrained immune reactivity. In this setting, control of the immune response did not seem to emerge from reduced T cell priming or antigen-specific T cell deletion as comparable frequencies of IFNγ-producing effector CD8⁺ and CD4⁺ T cells were found in K14-mOVA lymph node and wild-type lymph node recipients, upon in vitro re-stimulation of splenic cells with OVA peptides (% of IFNγ⁺ CD8⁺ and CD4⁺ T cells in wild-type lymph node transplant recipients vs K14-mOVA lymph node transplant recipients: 0.23 ± 0.08 vs 0.37 ± 0.17, p = 0.28; 0.28 ± 0.09 vs 0.55 ± 0.22, p = 0.47).

To address whether self-antigen presentation by lymph node stromal cells could have clinical relevance, we used a dual transplantation system in which we assessed whether transplantation of K14-mOVA lymph nodes would improve K14-mOVA skin graft acceptance. In these experiments, wild-type age-matched recipients were left untreated or were transplanted with either wild-type or K14-mOVA lymph nodes at week -4, followed by a skin transplant at day 0, after which they were monitored for 4 additional weeks. As compared to wild-type skin grafts, K14-mOVA skin grafts showed increased thickness and rigidity (not depicted) and an overall increased CRM score (*Figure 6B*), suggesting an ongoing process of rejection. Expression of CRM genes was significantly reduced, by more than 50%, by prior transplantation of K14-mOVA, but not wild-type lymph nodes (*Figure 6B* and *figure 6—figure supplement 1*), thus confirming the regulatory properties of self-antigen-expressing lymph node stromal cells. Remarkably, in contrast to K14-mOVA skin transplants, K14-mOVA lymph node transplants showed unaltered CRM scores when compared to wild-type transplants or to endogenous non-transplanted wild-type lymph nodes (*Figure 2—figure supplement 5*), which suggest that lymph node stromal cells are specifically endowed with regulatory potential. As assessed by the CRM, this capacity seemed to be, at least partially, dependent on MHC-II expression (*Figure 2—figure supplement 5*). Taken together, our data suggest that antigen-presenting lymph node stromal cells constrain immune responses in vivo independently of antigen-specific T cell deletion.

## Discussion

The last years have been instrumental in uncovering the crucial role of non-hematopoietic lymph node stromal cells in the maintenance of immune tolerance. Lymph node stromal cells have been shown to regulate the proliferation of activated T cells by inhibiting T cell homotypic interactions and restraining entry into cell cycle (*Lukacs-Kornek et al., 2011*; *Siegert et al., 2011*). In addition, they were shown to induce the deletion of auto-reactive CD8⁺ T cells via their capacity to produce, process, and present PTAs on MHC-I molecules (*Lee et al., 2007*; *Nichols et al., 2007*; *Magnusson et al., 2008*;

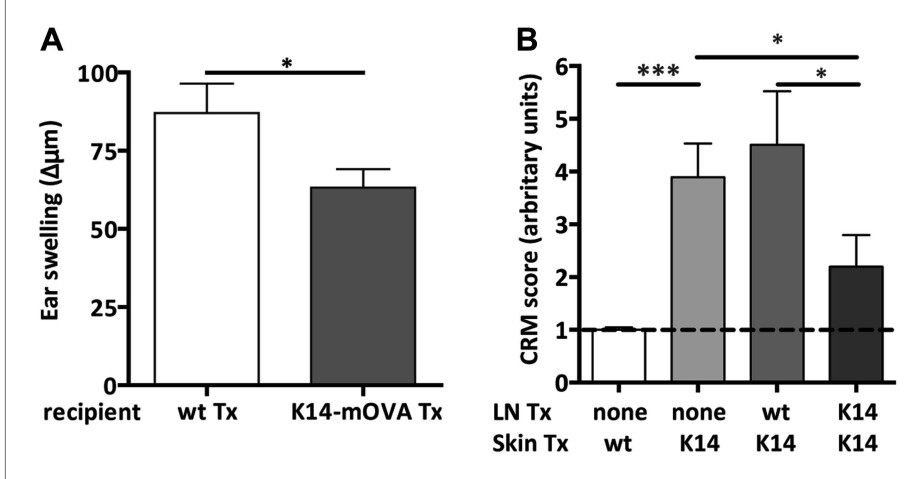

**Figure 6**. Endogenous OVA presentation by lymph node stromal cells constrains immune reactivity. (**A**) Wild-type mice transplanted with either wild-type (wt Tx) or K14-mOVA transgenic (K14-mOVA Tx) lymph nodes were immunized with OVA in incomplete Freund's adjuvant (IFA) in the tail base and re-challenged with OVA alone in both ears. In vivo delayed-type hypersensitivity (DTH) responses were determined by ear swelling. The data represent mean ± SEM; n = 5 mice per group; *p ≤ 0.05. (**B**) Wild-type mice either left untreated or transplanted with wild-type or K14-mOVA transgenic lymph nodes, at week -4, were transplanted with either wild-type or K14-mOVA skin on day 0. 4 weeks after skin transplantation, skin grafts were isolated and mRNA transcripts belonging to the common rejection module (CRM) analyzed by real-time PCR. The aggregate CRM score is shown. n = 6 mice per group; *p < 0.05, ***p ≤ 0.001.

The following figure supplement is available for figure 6:

**Figure supplement 1**. Self-antigen-presenting stromal cells constraint immune reactivity in a dual transplantation system.

*Cohen et al., 2010*; *Fletcher et al., 2010*) and to induce CD4$^+$ T cell dysfunction via acquired peptide–MHC-II complexes (*Dubrot et al., 2014*). In the present study, we have investigated the role of these cells as antigen-presenting cells via endogenous, non-acquired MHC-II. We have shown that MHC-II-mediated antigen presentation by lymph node stromal cells promotes the maintenance of Tregs contributing to the preservation of immune quiescence.

MHC-II molecules were expressed on the surface of FRCs, LECs, and BECs and endowed these cells with the ability to control immune reactivity. As evidenced by transplantation of MHC-II KO lymph nodes, which resulted in selective MHC-II deficiency on the transplanted lymph node stromal cell compartment, MHC-II expression on lymph node stromal cells controlled the activation of CD4$^+$ as well as CD8$^+$ T cells. Both T cell subsets showed an increased frequency of CD62L$^-$CD44$^+$ activated cells in MHC-II KO lymph node transplants as compared to wild-type lymph node transplants. These results are particularly relevant, as in the context of the ability of lymph node stromal cells to capture dendritic cell-derived peptide–MHC-II complexes (*Dubrot et al., 2014*), they reveal a critical role for lymph node stromal cell endogenously expressed MHC-II in immune regulation. The effect of stromal cell MHC-II deficiency on CD8$^+$ T cell activation was rather unexpected as MHC-II molecules are not thought to mediate cellular interactions with CD8$^+$ T cells. Additional experiments, however, showed that this effect resulted from the lack of regulation by CD4$^+$ T cells. In line with this, we found that in the absence of endogenous MHC-II expression on lymph node stromal cells, the maintenance as well as the proliferation of Tregs was impaired. Together, these data showed that for the homeostatic regulation of Tregs, similar as for other lymphocyte populations (*Cyster et al., 1994*; *Dummer et al., 2001*), access to the lymph node parenchyma is required. Importantly, our data also suggest that the maintenance of Tregs and their control of T cell activation are spatially linked and must occur within the same location. In fact, even though Treg numbers were adequately maintained in secondary lymphoid organs other than the transplant, the inability to maintain them within MHC-II KO lymph node transplants resulted in the local activation of T cells, which in the long-term resulted in the rejection

of the MHC-II KO transplants. This is significant as it suggests that, in order to minimize host vs graft disease in transplanted individuals, local maintenance of Tregs should be enforced.

Of significance, as ETACs persisted and still expressed MHC-II in MHC-II KO lymph node transplants, we can conclude that, despite their ability to inactivate CD4+ and CD8+ T cells (*Gardner et al., 2008*, *2013*), these cells cannot fully compensate for the lack of lymph node stromal cell-imposed regulation.

One could argue that rejection of MHC-II-deficient transplants could result from the introduction of MHC-I defects or unappreciated antigens during genetic targeting to generate MHC-II KO mice. In our experiments, we used MHC-II KO animals that were established by the insertion of a hygromycin-resistance cassette at the deletion site, spanning from the second exon of H2-Ab1 to the third exon of the H2-Ea gene, into 129S2/SvPas-derived H1 embryonic stem cells (*Madsen et al., 1999*). As the H2 locus of C56BL/6J and 129Sv strains is identical (*b* haplotype) and our MHC-II KO animals show normal MHC-I expression (*Madsen et al., 1999*), rejection of MHC-II KO lymph nodes could not have been caused by impaired recognition of MHC-I molecules. Indeed, if faulty recognition of MHC-I molecules would be responsible for MHC-II KO transplant rejection, one would not expect CD8+ T cell activation to be present and augmented by CD4+ T cell deletion. Unappreciated antigens derived from the hygromycin-resistance cassette could play a role in MHC-II KO lymph node rejection, as recipient mice had never been exposed to such antigens. However, previous experience with lymph nodes harboring GFP constructs (*Molenaar et al., 2009*) and presently also with the K14-mOVA transgene suggest that possibility to be remote. Taken these considerations as a whole, our data strongly support the notion that endogenous MHC-II expression on lymph node stromal cells is critical for maintaining low CRM scores and thus safeguarding tolerance.

Treg development in the thymus occurs through agonist selection on MHC-II presented peptides (*Josefowicz et al., 2012*). Similarly, our data with the K14-mOVA transgenic lymph node transplantation and OT-II T cell transfer system confirmed that the peripheral maintenance of the Treg pool required MHC-II-mediated presentation of endogenous antigens as well. Presentation of OVA-derived peptides by the transplanted lymph node stromal cell compartment led to enhanced numbers of OT-II Tregs within the lymph node transplant, which was particularly evident for CD62L-expressing CD4+Foxp3+ Tregs. Determination of the origin of the expanded cells warrants further research. Given that a large fraction of the expanded Treg population expressed Helios (*Figure 5—figure supplement 2*), a transcription factor originally associated with Treg development in the thymus (*Thornton et al., 2010*), it may seem that transferred thymus-derived OT-II Tregs were specifically maintained via cognate interactions with the K14-mOVA lymph node stroma. Alternatively, as Helios expression was more recently shown to be induced upon T cell activation (*Akimova et al., 2011*) preceding Foxp3 induction on peripherally induced Tregs (*Gottschalk et al., 2012*), it may be that our expanded Treg population reflects peripheral differentiation of naïve OT-II T cells into OT-II Tregs. In either case, the increase in OT-II Tregs did not involve cellular proliferation. This contrasts with the effect of self-antigen recognition in peripheral tissues, such as the skin, which induces vigorous Treg proliferation (*Rosenblum et al., 2011*). Overall, our in vitro and in vivo data suggest that the antigen-mediated interaction between lymph node stromal cells and Tregs provides specific survival signals to the latter cells that may allow antigen-stimulated Tregs to outcompete Tregs that have not seen their cognate antigen. If present, such a mechanism would most likely select the Treg repertoire to match the peripheral need for immune regulation. Supporting this hypothesis, it was previously shown that the peripheral Treg repertoire differs significantly between different anatomical locations (*Lathrop et al., 2008*), a situation that may reflect differences in regional lymph nodes (*Wolvers et al., 1999*; *Hammerschmidt et al., 2008*).

Transplantation of K14-mOVA transgenic lymph nodes was associated with in vivo development of OVA unresponsiveness. Importantly, in contrast with previous reports showing that PTA expression by lymph node stromal cells drives the deletion of self-reactive CD8+ T cells (*Lee et al., 2007*; *Nichols et al., 2007*; *Gardner et al., 2008*; *Magnusson et al., 2008*; *Cohen et al., 2010*; *Fletcher et al., 2010*), in our system OVA unresponsiveness did not seem to be related to this mechanism. We observed comparable frequencies of OVA-specific IFNγ-producing CD8+ T cells between mice transplanted with K14-mOVA transgenic lymph nodes and mice transplanted with wild-type lymph nodes. Unresponsiveness did not seem to arise from the deletion of OVA-specific IFNγ-producing CD4+ T cells either. This is in agreement with the recent observation by Magnusson et al., showing that self-antigen presentation by lymph node stromal cells does not promote the deletion of self-reactive CD4+ T cells (*Magnusson et al., 2008*). This is however, in contrast with the findings that LECs induce

antigen-specific CD4$^+$ T cell apoptosis (*Dubrot et al., 2014*). The reasons for such disparate results are currently unknown, but may relate to the use of different experimental systems and/or to putative differences in self-antigen processing by lymph node stromal cells and dendritic cells. Regardless of the origin of such discrepancies, together these results highlight the existence of multiple regulatory mechanisms that in concert safeguard the homeostasis of the immune system.

Importantly, as lymph node stromal cells enwrap the tubular system that connects the incoming afferent lymphatic vessels with the inner core of lymph nodes (*Sixt et al., 2005*), it will be important to assess in the future whether exogenous antigens can be taken up from the lymph node conduits by lymph node stromal cells and presented to T cells. This may be particularly relevant in the context of immune responses as presentation of exogenous antigen by lymph node stromal cells may contribute to reactive CD8$^+$ T cell deletion, to CD4$^+$ T cell dysfunction, and/or to Treg expansion and therefore prevent excessive inflammation or contribute to the contraction of the immune response. Alternatively, as lymph node stromal cells express toll-like receptors (TLRs) (*Fletcher et al., 2010*), it may be that during immune responses antigen processing gets redirected (*Blander and Medzhitov, 2006*) and that lymph node stromal cells, like dendritic cells (*Reis e Sousa, 2006*), mature from a tolerogenic phenotype towards an immunogenic one.

In conclusion, we showed a hitherto unrecognized role for lymph node stromal cells in the maintenance of peripheral Tregs and immune quiescence. MHC-II-mediated antigen presentation by lymph node stromal cells was essential for the steady-state maintenance of Tregs as well as for their homeostatic recovery in lymphocyte-depleted environments. Tregs, in turn, prevented immune activation, which was required for graft acceptance. Importantly, in contrast with previous reports (*Lee et al., 2007*; *Nichols et al., 2007*; *Magnusson et al., 2008*; *Cohen et al., 2010*; *Fletcher et al., 2010*), in vivo immune regulation by self-antigen-expressing lymph node stromal cells in our transplantation setting was not related to self-reactive T cell deletion, suggesting that lymph node stromal cells may use multiple mechanisms to control self-reactivity.

## Materials and methods

### Mice

C57BL/6J (wild-type), C57BL/6.129S2-H2$^{delAb1-Ea}$/J (MHC-II KO), C57BL/6-*Rag2*$^{tm1Cgn}$/J (*Rag2*$^{-/-}$), and human keratin 14 membrane-bound ovalbumin (K14-mOVA), C57BL/6-Tg(TcraTcrb)1100MjB/J (OT-I), and C57BL/6-Tg(TcraTcrb)425Cbn/J(OT-II) transgenic mice were kept at the Vrije University Medical Center animal facility under SPF conditions. All animal experiments were reviewed and approved by the Vrije University Scientific and Ethics Committees.

### Lymph node transplantation

Lymph node transplants were performed as previously described (*Mebius et al., 1993*). Briefly, wild-type or *Rag2*$^{-/-}$ recipient mice were anaesthetized with xylazine and ketamine and their popliteal lymph nodes removed and replaced by peripheral (axillary, brachial, or inguinal) lymph nodes of donor origin (wild-type, MHC-II KO, or K14-mOVA). Each recipient mouse received two identical lymph nodes. Transplants were allowed to reconnect to the blood and lymphatic vasculatures for at least 4 weeks, upon which their function was tested.

### Skin transplantation

Wild-type recipient mice were anaesthetized with xylazine and ketamine and shaved on the flank, after which a skin flap of 1 cm$^2$ was removed. Back skin from shaved, sex-matched wild-type or K14-mOVA donor mice was removed and cut into 1 cm$^2$ pieces, which were sutured to the recipients' skin.

### In vivo T cell depletion

In lymph node transplant recipients of wild-type origin, depletion of CD4$^+$ T cells was achieved by intraperitoneal treatment with the anti-CD4 antibody GK1.5. Treatment was renewed twice per week starting 1 week before the performance of lymph node transplants until the end of the experiments. The first two injections contained each 200 µg of antibody, whereas all the remaining only 100 µg. In *Rag2*$^{-/-}$ recipient mice, homeostatic expansion of T cells co-transplanted with the donor lymph nodes was prevented by treatment with the anti-CD4 and anti-CD8 antibodies GK1.5 and 2.43. Each mouse received 3 intraperitoneal injections of 200 µg of each antibody evenly distributed during the first week after transplantation.

## Cell isolation, labeling, and transfer

To obtain single cell suspensions for in vivo transfer, lymph nodes and spleens of wild-type and OT-II transgenic mice were squeezed through cell strainers. Erythrocytes were lysed with ammonium-chloride–potassium (ACK) lysis buffer. OT-II T cells were isolated with the CD4$^+$ cell negative isolation kit from Miltenyi (Leiden, The Netherlands) following the manufacturer's instructions—purity was on average >85%. CFSE (Invitrogen, Breda, The Netherlands) labeling was performed as previously described (*Wolvers et al., 1999*). Briefly, cells were resuspended at $40 \times 10^6$ cells/ml and incubated with 5 µM CFSE (Molecular Probes, Breda, the Netherlands) for 10 min at 37°C. $10^7$ CFSE-labeled cells were transferred into the tail vein of transplanted mice.

## Delayed-type hypersensitivity (DTH) response

4 weeks after lymph node transplantation, transplanted mice were immunized with 100 µg OVA (Sigma-Aldrich, St Louis, MO) in 25 µl incomplete Freund's adjuvant (Sigma–Aldrich) plus 25 µl PBS (B. Braun, Euterpehof, The Netherlands) subcutaneously in the tail base. 5 days later, the immunized mice were injected intradermally in both ears with 10 µl of PBS containing 10 µg of OVA. Ear thickness was measured with a micrometer (Mitutoyo, Tokyo, Japan) before and 24 hr after secondary challenge in a blinded fashion. Ear swelling was calculated as the difference in ear thickness at 0 and 24 hr. Each mouse provided two independent measurements (two ears) that were averaged to determine the average ear swelling. *Ex vivo* evaluation of effector T cells was performed 1 day after the last ear measurement. Briefly, lymphocytes were isolated from spleens and re-stimulated with the OVA peptides SIINFEKL (OVA$_{257-264}$—0.2 µg/ml) and EKLTEWTSSNVMEER (OVA$_{265-279}$—200 µg/ml) for 24 hr. The last 4 hr of culture were performed in the presence of Golgiplug (BD Biosciences, Breda, The Netherlands).

## Primary lymph node stromal cell sorting and flow cytometry

Peripheral lymph node single cell suspensions for cell sorting and flow cytometry were obtained by lymph node enzymatic digestion with 0.2 mg/ml collagenase P (Roche, Penzberg, Germany), 0.8 mg/ml dispase II (Roche), and 0.1 mg/ml DNAse I (Roche) as described by *Fletcher et al. (2011)*. Prior to cell sorting, the single cell preparation was enriched for non-hematopoietic stromal cells by negative selection of hematopoietic cells using the α-mouse CD45 (clone MP33) PE.Cy7 antibody from eBioscience (Halle-Zoersel, Belgium) and the mouse PE selection kit from StemCell Technologies (Grenoble, France), accordingly to the manufacturer's instructions. Surface stainings were performed on ice for 30 min in PBS/2%FCS/5 mM EDTA. Intracellular stainings were performed in permeabilization buffer, upon fixation in fixation/permeabilization buffer (both from eBioscience) for 30 min. The antibodies used were: α-CD31 (clone ERMP12) and α-MHC-II (M5/114) labeled with AlexaFluor 555 and 488 (Invitrogen), respectively; unlabeled hamster α-mouse gp38 (8.1.1) developed with goat anti-hamster AlexaFluor 647 (Invitrogen); biotin-conjugated α-mouse CD45.1 (A20; eBioscience) and α-mouse TER-119 (eBioscience) developed with streptavidin AlexaFluor 488 (Invitrogen) and PE.Cy7 (eBioscience), respectively; α-CD3 (17A2) eFluor 660, α-CD4 (GK1.5) AlexaFluor 488, PerCP.Cy5 and PE.Cy7, α-CD8 (53-6.7) PE.Cy7 and APC.eFluor 780, α-CD11c (N418) APC, α-CD25 (PC61.5) PE, α-CD44 (IM7) PE, α-CD45 (MP33) PE.Cy7 and APC.Cy7, α-CD62L (MEL-14) PE.Cy7, α-EpCAM (G8.8) PE, α-FoxP3 (FJK-16s) AlexaFluor 647, and α-IFNγ (XMG1.2) APC purchased from eBiosciences; and α-Helios (22F6) PacificBlue obtained from Biolegend (Fell, Germany). Live and dead cells were discriminated either with 7AAD, SytoxBlue, or live/dead Near-IR fixable dead cell stain (all from Invitrogen). Cells were sorted on a MoFlo sorter (DakoCytomation, Heverlee, Belgium) or analyzed on either a Cyan ADP (DakoCytomation) or a CantoII (BD Biosciences) flow cytometer with FlowJo software (TreeStar, Ashland, OR). Fluorescence minus one (FMO) and isotype control stain sets were used to assess detection thresholds.

## Immunofluorescence

Lymph node transplants were embedded in OCT compound (Sakura Finetek, Leiden, the Netherlands) and snap frozen in liquid nitrogen. Frozen blocks were cut into 7-µm sections. Sections were stained by incubation with the relevant antibodies for periods of 45 min at room temperature; and when needed, further incubated with appropriate secondary antibodies/reagents for 30 min. The antibodies used were: α-B220 (6B2), α-CD31 (ERMP12), α-MHC-II (M5/114), and Moma2 labeled in house with either Alexa Fluor 647 or AlexaFluor 555 (Invitrogen); unlabeled rat anti-mouse ERTR7 developed with

anti-rat AlexaFluor 488 (Invitrogen); unlabeled rabbit anti-mouse collagen type I (polyclonal; AbCAM, Cambridge, UK) and anti-mouse CSFR1 (polyclonal, Sigma–Aldrich, St Louis, MO) developed with anti-rabbit AlexaFluor 647 and anti-rabbit AlexaFluor 555 (Invitrogen), respectively; biotin-labeled α-mouse CD11c (N418; BioLegend, Fell, Germany) developed with streptavidin conjugated to AlexaFluor488 (Invitrogen); and directly labeled α-CD3 (17A2) eFluor 660, α-F4/80 (BM8) eFluor 660, and α-Lyve-1 (ALY7) AlexaFluor 488 (all from eBioscience). Pictures were taken on a DM6000 Leica immunofluorescence microscope (Leica Microsystems, Rijwijk, the Netherlands). Analysis of the area occupied by CD11c$^+$CSFR1$^+$ clusters was performed with ImageJ.

## In vitro generation of lymph node stromal cell lines

Peripheral lymph node single cell suspensions for culture purposes were obtained by enzymatic digestion as described above. To obtain enriched primary stromal cell cultures, the resulting cell suspensions were plated on collagen-coated flasks and cultured in RPMI (Invitrogen) containing 10% heat-inactivated FCS (Gibco, Breda, The Netherlands), 2% glutamine (Lonza, Basel, Switzerland), 2% penicillin–streptomycin (Lonza), and 50 μM 2-mercapethanol (Merck, Haarlen, The Netherlands). Stromal cells were allowed to adhere to the collagen matrix overnight, after which non-adherent immune cells were washed away. Long-term culture and regular fractioning of these cultures permitted the establishment of immortalized lymph node stromal cell lines that were subsequently sorted and repeatedly characterized by flow cytometry to ensure the maintenance of stable phenotypes and the absence of CD45$^+$ hematopoietic cells.

## In vitro T cell and lymph node stromal cell co-cultures

To isolate T cells for in vitro culture, lymph nodes and spleens from wild-type, OT-I, and OT-II mice were dissected and squeezed through cell strainers. Erythrocytes were lysed with ammonium-chloride–potassium (ACK) lysis buffer. CD4$^+$ wild-type and OT-II T cells were magnetically isolated with the CD4$^+$ cell negative isolation kit and CD8$^+$ OT-I T cells with the CD8$^+$ cell negative isolation kit (both from Miltenyi), following the manufacture's protocols. To determine T cell activation/proliferation, T cells were cultured together with lymph node stromal cell lines derived from K14-mOVA mice in the presence of 200 U/ml mouse recombinant IL2 (BioLegend). As a positive control for T cell activation/proliferation, T cells were cultured with CD3/CD28 T cell expander Dynabeads (Invitrogen) at a 1/1 ratio. To assess the role of lymph node stromal cells in antigen-specific Treg maintenance, T cells were cultured together with lymph node stromal cell lines derived from K14-mOVA mice in the absence of recombinant IL2. In some experiments, the MHC-II blocking antibody M5/114 (10 μg/ml) was added to the co-cultures.

## RNA isolation, complementary DNA (cDNA) synthesis, and real-time PCR

To analyze gene expression in transplanted lymph nodes and skin, transplanted tissues were dissected and immediately stored in Trizol (Gibco) at −80°C. Tissue homogenization was performed with an Ultra-Turrax T10 disperser (IKA, Staufen, Germany). mRNA was isolated and cDNA synthesized, as previously described (*Baptista et al., 2013*).To analyze the transcript expression in sorted cells, mRNA was isolated with the mRNA capture kit from Roche and cDNA synthesized with the Reverse Transcription System from Promega (Leiden, The Netherlands), according to the manufacturers' instructions. In both cases, real-time PCR was performed on StepOne real-time PCR systems from Applied Biosystems (Bleiswijk, The Netherlands). The expression of each transcript was analyzed and normalized for the expression of selected housekeeping genes with geNORM v3.4 software (Center for Medical Genetics, Ghent University Hospital, Ghent, Belgium).

## Data analysis and statistics

T cell proliferation was analyzed with FlowJo software (Tree Star). Division indexes represent the average number of cell divisions that a cell in the original population has undergone according to the formula: division index = total number of cell divisions/total number of precursor cells present at the beginning of the assay. The common rejection module (CRM) score represents the geometric mean expression of the genes comprised in the CRM (*Khatri et al., 2013*). Statistical analysis was performed with GraphPad Prim v.4 (GraphPad Software, San Diego, CA). One-way non-parametric analysis of variance (Kruskal–Wallis test) coupled to Dunn's or Tukey's multiple comparison tests, non-parametric repeated measures analysis of variance (Friedman's test) coupled to Dunn's multiple comparison tests, and non-parametric unpaired and paired t-tests were used as required. Data in *Figure 6B* were subjected to a Box–Cox transformation, to approximate it to a normal distribution,

after which one-way Anova statistical analysis was performed on the transformed data, as described above. All data is expressed as mean ± SEM. p-values <0.05 were considered significant.

## Acknowledgements

We thank Wessel N van Wieringen (Dept. Epidemiology and Biostatistics, VUMC, Amsterdam, the Netherlands) for his help with statistical analysis.

This work was supported by grants from Fundação para a Ciência e Tecnologia—Portugal (SFRH/BD/33247/2007 to APB)—and the Netherlands Organization for Scientific Research (VICI grant 918.56.612 to REM, ALW program grants 820.02.004 to RR, 823.02.011 to JJK, and 854.10.005 to GG, and VENI grant 916.13.011 to RMR).

## Additional information

### Funding

| Funder | Grant reference number | Author |
| --- | --- | --- |
| Nederlandse Organisatie voor Wetenschappelijk Onderzoek | VICI grant 918.56.612 | Reina E Mebius |
| Nederlandse Organisatie voor Wetenschappelijk Onderzoek | ALW program grant 820.02.004 | Ramon Roozendaal |
| Nederlandse Organisatie voor Wetenschappelijk Onderzoek | ALW program grant 823.02.011 | Jasper J Koning |
| Nederlandse Organisatie voor Wetenschappelijk Onderzoek | VENI grant 916.13.011 | Rogier M Reijmers |
| Fundação para a Ciência e a Tecnologia | PhD Scholarship SFRH/BD/33247/2007 | Antonio P Baptista |
| Nederlandse Organisatie voor Wetenschappelijk Onderzoek | ALW program grant 854.10.055 | Gera Goverse |

The funders had no role in study design, data collection and interpretation, or the decision to submit the work for publication.

### Author contributions

APB, RR, Conception and design, Acquisition of data, Analysis and interpretation of data, Drafting or revising the article; RMR, JJK, Acquisition of data, Analysis and interpretation of data, Contributed unpublished essential data or reagents; WWU, MG, EDK, RM, GG, MMSS, Acquisition of data, Analysis and interpretation of data; JMMH, Provided MHC-II KO mice, Analysis and interpretation of data, Drafting or revising the article; MB, Provided K14-mOVA mice, Analysis and interpretation of data, Drafting or revising the article; REM, Conception and design, Analysis and interpretation of data, Drafting or revising the article

### Ethics

Animal experimentation: All animal experiments were reviewed and approved by the Vrije University Scientific and Ethics Committees (protocols MCB09-35, MCB10-01 and MCB13-06). All surgery was performed under xylazine and ketamine anesthesia.

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
