## [Decision Letter]

Thank you for sending your work entitled “Lymph node stromal cells constrain immunity via MHC class II self-antigen presentation” for consideration at *eLife*. Your article has been favorably evaluated by Tadatsugu Taniguchi (Senior editor), a guest Reviewing editor, and 3 reviewers, one of whom, Victor Engelhard (reviewer #2), has agreed to reveal his identity.

The Reviewing editor and the other reviewers discussed their comments before we reached this decision, and the Reviewing editor has assembled the following comments to help you prepare a revised submission.

Your paper is of definite interest to the editors and the reviewers. There were a number of important issues that were raised. We would like to proceed with your paper provided that you are willing to consider and answer many of their comments. The following are issues that you should consider of importance and that will require further analysis.

Although you need to consider the various issues raised by the three reviewers there are some major ones to pay attention to.

For one, the examination of MHC-II expression on the lymph nodes was brought up by the three reviewers and it is clearly one that you need to examine at depth. Note comment 1 of Reviewer 1 and Reviewer 2, and the comment regarding Figure 1 from Reviewer 3.

It is also important that you give us your interpretation of Reviewer 2 #5 concerning the possible transfer of pMHC-II complexes from one cell to another. This is also brought up by Reviewer 3 in the comment on Figure 1. Please address this important point including new experiments so as to clarify this issue.

We trust that you will be able to evaluate the reviewers’ comments. And that we can see a revised version of your interesting paper.

*Extract from Reviewer 1*:

Figure 5 shows an increase in the % and number of Tregs in subjects that received K14-OVA lymph node transplants compared with WT lymph node transplants. Figure 6 shows that K14-OVA expression in transplanted lymph nodes reduces the DTH response (as measured by ear swelling and CRM score) relative to OVA-free control transplanted lymph nodes. However, no analysis of Tregs is provided in this part of the study. The data in Figures 5 and 6 suggest that some form of immunoregulation is conferred by OVA expression in the transplanted lymph node however both donor and host tissues express MHC class II in these experiments. OVA expressed in the donor lymph node could have been presented by either donor lymph node cells (stroma), through the direct antigen presentation pathway, or by host antigen presenting cells that enter the graft (dendritic cells, monocytes) and present via an indirect pathway (endocytosis or phagocytosis of OVA expressing cells or via membrane transfer). Ideally, these transplant experiments would have been done in hosts with MHC class II deficiency in bone marrow derived antigen presenting cells such that the only MHC class II expressing cells would come from the donor tissue. The requirement for MHC class II in CD4 T cell and Treg development however makes it impossible for the authors to do these experiments using the MHC class II global knockout mice at hand. However, a definitive analysis of the cells presenting MHC class II-OVA peptides and Treg maintenance in their experimental system would be necessary to substantiate the authors' claims and conclusions. Putting these issues aside, the critical comparison in Figure 6 shows an interesting trend in the CRM score when transferred lymph node contains cells expressing OVA but importantly this difference does not reach statistical significance and therefore does not support the claims made by the authors.

Additional points:

1) The MHC class II analysis in Figure 1 should be performed transplanted lymph nodes and extended to include various subsets of bone marrow derived antigen presenting cell types in a manner where the donor and host cells can be parsed. This is critical as it allows the reader to assess precisely which cell subsets lack MHC class II within the transplanted lymph node.

2) The images in Figure 2 are provided to show structural damage to the transplanted lymph node when the donor tissue lacks MHC class II. However, this data cannot support the authors' claim without providing images for the control tissues (transplanted lymph node when the donor tissue is MHC class II+). In addition, quantification of the structural damage depicted in the images should be provided.

3) In two separate places (Figure 1–figure supplement 1; Figure 2—figure supplement 4) data for ETAC-like cells were included in the Figures but not described in the manuscript until the Discussion.

4) Characterization including phenotype and purity of the stromal cell lines (both control and K14-derived) must be provided. Control and antigen-expressing stromal cell lines should belong to the same lineage.

5) A major point of this paper is that lymph node stromal cells support Treg maintenance however the data in Figure 4 indicate that the number of Tregs that persist in a culture with lymph node stromal cells is extremely low. How does this efficiency compare with dendritic cells? Representative flow cytometry data should accompany the graph in Figure 4.

6) Flow cytometry data for the efficiency of CD4 or CD8 T cell depletion in the donor tissues must be shown.

7) Statistical analysis of data in Figure 2—figure supplement 2 must be shown.

8) Figure 3, the y-axis is missing the values for cell counts.

*Reviewer 2 (pay particular attention to #1)*:

1) There is no characterization of cell population distributions and MHC-II expression patterns in transplanted WT LN or MHC-IIko LN. These are essential to understand the differences in T cell behavior in these two LN, and to support the assertions in the Results section and the significance of Figure 2—figure supplement 1.

2) What is the basis for rejection of MHC-IIneg LN? The analyses of T cells in an MHC-IIneg LN are in an environment of inflammation and tissue damage, raising the question of whether their characteristics are causing the pathology or are affected by it, apart from any influence of Treg.

3) The authors conclude that lack of MHC-II molecules on LNSC leads to diminished antigen-specific Treg maintenance/function and enables more robust activation of conventional CD4 and CD8 T cells. An alternative is based on the observation that MHC-II molecules are antigen-nonspecific ligands for the LAG-3 inhibitory receptor. This could affect the activation state of CD8 T cells and non-regulatory CD4 T cells independently of any influence on Treg representation. The data in right panels of Figure 2 and Figure 2—figure supplement 3 comparing WT and MHC-IIko LN in GK1.5 treated mice are strongly suggestive of this.

4) Long-term LNSC lines are problematic, as the lineages have different survival capabilities and often lose differentiated characteristics. The authors acknowledge that their ova negative line is FRC like, while the ova pos line is LEC like. Thus, comparing their effects on T cells to establish a role for antigen (Figure 4) is very problematic. In Figure 4, even OVAneg LNSC lines are associated with a substantial representation of FoxP3+ cells in both bulk CD4 and OT-II T cells. Based on Figure 4—figure supplement 1, this is not due to proliferation, and is associated with upregulation of CD25 in CD8 T cells, but not OT-II cells. The authors state that this is an increase in survival of FoxP3+ cells, but FoxP3neg cells should also be examined, and there should be a zero time control to establish whether antigen is associated with selective survival or differentiation. Regardless, the distinct phenotypes of the cell lines will make it difficult to definitively conclude that antigen expression is responsible.

5) The authors interpret the results in Figure 5 as though only LNSC present antigen, but do not account for the possibility that ova positive LNSC transfer antigen to MHC-II positive hematopoetic cells in the same LN, or transfer preformed MHC-II antigen complexes. Thus, the results do not unambiguously establish that LNSC are responsible for any increase in representation of Treg or Treg subsets via direct presentation of antigen.

6) Figure 5 and Figure 5—figure supplement 1 show several instances in which changes in absolute numbers of Treg or Treg subsets change, but frequency or representation does not - or the converse. This must mean that other cell populations are also changing in response to characteristics of the transplanted LN. This needs to be clarified, as the manuscript text indicates that LN stroma contribute to the selective maintenance of antigen-specific.

*Reviewer 3 (comment on*
Figure 1
*needs to be prioritized)*:

1) Figure 1: the authors claim a “selective absence of MHCII on LNSC from Tx MHCII-/- mice (transplanted in WT)”. Given the recent publication showing that MHCII are not only endogenously expressed by LNSC, but also acquired from DC, this result is quite surprising. It would be mandatory to show this result and compare with the situation in which DC do not express MHCII. For instance, show an absence of MHCII expression by LNSC from MHCII-/- LN Tx into chimeric mice (MHCII-/- BM into irradiated WT). Importantly, can the authors demonstrate any difference in Treg proliferation in the context of MHCII-/- LN Tx compared to endogenous distal LN?

2) Figure 2 shows that T cell activation is increased in MHCII-/- LN Tx. As a demonstration of an effect happening locally in Tx LN, it is important to provide not only WT LN Tx, but distal endogenous LN fron MHCII-/- LN Tx mice as well.

3) Figure 3 shows T cell (Treg and non Treg) proliferation 48h after OTII transfer. Is the observed effect still happening at later time points? Here as well, a distal endogenous LN should be provided. In addition, the division index ratio is quite confusing, since we do not have any indication of the proliferation rate of the cells. Please provide % of T cell (Treg and non Treg) proliferation, as well as % of Tregs. Moreover, dendritic cells were shown to be responsible for homeostatic Treg proliferation (Nussensweig's lab). How do the authors explain their results in that context? If, as shown, there are less Tregs in MHCII-/- LN Tx, we would have expected increased non Treg T cell proliferation, whereas opposite results are observed. Can the authors comment on that?

4) Figure 4: The cell lines obtained and used by the authors (“FRC like OVAneg” and “LEC like OVApos”) represent a quite artificial system and are poorly relevant in the context of the paper (most experiments were performed in vivo). It is true that primary LEC / FRC cultures are contaminated by hematopoietic cells. However, in case the authors think these in vitro experiments are relevant for the paper, they need to sort LECs and FRC from those primary cultures to repeat the experiments. Furthermore, it is quite surprising that no CD8+ T cell proliferation was observed in these in vitro settings, since many studies have described early and transient CD8 T cell proliferation (followed by T cell deletion) after culture with by MHCI-Ag presenting LNSC.

5) Figures 5 and 6 are elegant and nice. The major issue I have is to understand whether the authors want to claim a local or a systemic effect of LNSC in dampening self-reactive T cell responses. Indeed, Figure 1 shows a local effect, whereas Figures 5 and 6 show a more general effect. Figure 5 shows a significant Treg increase in endogenous distal LN as well (although discrepancies exist between FACS dot plots (1.17% vs 1.7%) and histograms (about 3% vs >4% Treg). The authors need to clarify this point and, in case they think the effect is systemic, explain how it might work.

---

## [Author Response]

Extract from Reviewer 1:

Figure 5 shows an increase in the % and number of Tregs in subjects that received K14-OVA lymph node transplants compared with WT lymph node transplants. Figure 6 shows that K14-OVA expression in transplanted lymph nodes reduces the DTH response (as measured by ear swelling and CRM score) relative to OVA-free control transplanted lymph nodes. However, no analysis of Tregs is provided in this part of the study. The data in Figures 5 and 6 suggest that some form of immunoregulation is conferred by OVA expression in the transplanted lymph node however both donor and host tissues express MHC class II in these experiments. OVA expressed in the donor lymph node could have been presented by either donor lymph node cells (stroma), through the direct antigen presentation pathway, or by host antigen presenting cells that enter the graft (dendritic cells, monocytes) and present via an indirect pathway (endocytosis or phagocytosis of OVA expressing cells or via membrane transfer). Ideally, these transplant experiments would have been done in hosts with MHC class II deficiency in bone marrow derived antigen presenting cells such that the only MHC class II expressing cells would come from the donor tissue. The requirement for MHC class II in CD4 T cell and Treg development however makes it impossible for the authors to do these experiments using the MHC class II global knockout mice at hand. However, a definitive analysis of the cells presenting MHC class II-OVA peptides and Treg maintenance in their experimental system would be necessary to substantiate the authors' claims and conclusions. Putting these issues aside, the critical comparison in Figure 6 shows an interesting trend in the CRM score when transferred lymph node contains cells expressing OVA but importantly this difference does not reach statistical significance and therefore does not support the claims made by the authors.

We agree with this reviewer that in the experiments shown in Figures 5 and 6 we were not able to distinguish whether direct antigen presentation by stromal cells or indirect antigen presentation, via host antigen presenting cells, is responsible for the increase in % and number of Treg cells and the reduction in DTH response and CRM score. However, together with the data presented in Figure 4, in which we show that antigen presentation by lymph node stromal cells in vitro leads to the selective survival of antigen-specific CD4+FoxP3+ Treg cells in a MHC-II-dependent manner, we have established that lymph node stromal cells can directly present antigens to T cells. We fully realize that the transfer of MHC-II-antigen peptide complexes from dendritic cells to lymph node stromal cells, as published by Dubrot et al (JEM, 2014) is a mechanism by which lymph node stromal cells can also regulate their antigen presenting function. This mechanism would however not play a significant role in these settings, as dendritic cells are circulating and host-derived cells (as also shown in Figure 2—figure supplement 1 and further explained under point 1) and thus are not expressing the K14-OVA transgene. Reversely, the possibility that MHCII-peptide complexes are transferred from donor-derived stromal cells to circulating dendritic cells is indeed a possibility, and may occur in addition to the direct presentation of antigen by stromal cells, as we have shown in Figure 4.

As this reviewer pointed that the data shown in Figure 6 was interesting but not statistically significant, and therefore not relevant, we consulted a statistician to more closely have a look at our statistical analysis. Importantly, after careful re-analysis, the statistician pointed out that our data was not distributed normally, and therefore should undergo a Box-Cox transformation, to convert it to a normal distribution. Subsequently, applying one-way Anova on the transformed data revealed that the differences between the animals transplanted with wild-type lymph nodes and K14mOVA skin and animals transplanted with K14mOVA lymph nodes and skin reached a p-value of 0.036, and thus statistically significantly different from each other. Moreover, an even less stringent correction is allowed according to the statistician, approaching the data in a one-tailed manner, resulting in p-value of 0.018 between these two groups. We feel that this data is very relevant for the manuscript, as this shows the consequences of antigen presentation by lymph node stromal cells. In the revised manuscript, we updated Figure 6 and the statistical analysis entry on the Materials and methods section. We have also acknowledged the statisticians’ help.

*Additional points*:

1) The MHC class II analysis in Figure 1 should be performed transplanted lymph nodes and extended to include various subsets of bone marrow derived antigen presenting cell types in a manner where the donor and host cells can be parsed. This is critical as it allows the reader to assess precisely which cell subsets lack MHC class II within the transplanted lymph node.

We agree with this reviewer that analysis of MHC-II expression in transplanted lymph nodes is of critical importance. Therefore, we now include such analysis in Figure 2—figure supplement 1. Whereas dendritic cells and extrathymic AIRE-expressing cells (ETACs) remain MHC-II positive (similar % of MHC-II+ cells within CD11c+ and EpCAM+ cells in wild-type and MHC-II-/- transplants), lymph node stromal cells exhibit a significant reduction in MHC-II expression in MHC-II-/- transplants as compared to wild-type transplants. MHC-II+ stromal cells in MHC-II-/- transplants are likely to have acquired MHC-II expression from dendritic cells as shown by Dubrot et al, JEM 2014. It is noteworthy to point out that gp38^-^CD31^-^ (double negative) cells do not express MHC-II even in wild-type intact animals and therefore contribute to the negative peak present in both wild-type and MHC-II-/- transplants.

2) The images in Figure 2 are provided to show structural damage to the transplanted lymph node when the donor tissue lacks MHC class II. However, this data cannot support the authors' claim without providing images for the control tissues (transplanted lymph node when the donor tissue is MHC class II+). In addition, quantification of the structural damage depicted in the images should be provided.

We provide the analysis requested by showing images from wild-type transplants in Figure 2—figure supplement 3. In addition we have quantified the amount of structural damage by assessing the area occupied by CD11c+CSFR1+ macrophage clusters in multiple sections. We express these data as percentage of total lymph node area.

3) In two separate places (Figure 1–figure supplement 1; Figure 2—figure supplement 4) data for ETAC-like cells were included in the Figures but not described in the manuscript until the Discussion.

In the course of our experiments, we analyzed EpCAM^+^ ETAC-like cells given the original publication stating that these cells were of stromal origin (Gardner et al. Science 2008), but found that they were most likely a subset of dendritic cells as they expressed high levels of CD45, CD11c and MHC-II. More importantly, these cells could be observed within MHC-II^-/-^ lymph node transplants and expressed normal MHC-II levels (see Figure 2—figure supplement 1). Together, these data suggested that ETACs were not contributing to the effects on T cell activation and lymph node rejection that we observed. We stated these data in the discussion and showed it in supplementary figures. In the revised version of our manuscript, we highlight these data in the Results section, as to clarify from early on that ETACs are likely not involved in the phenotypes described.

4) Characterization including phenotype and purity of the stromal cell lines (both control and K14-derived) must be provided. Control and antigen-expressing stromal cell lines should belong to the same lineage.

We agree with this reviewer that the stromal cell lines used in our experiments should have belonged to the same lineage. However, despite several attempts to generate large panels of stromal cell lines representing each lineage and of different genetic backgrounds, we have not managed to do so. Indeed, in our hands FACS-sorting of stromal cells immediately after their enzymatic isolation resulted in highly damaged cells that quickly died in culture. Thus, we have generated our cell lines by culture of unfractionated lymph node cells, which were allowed to adhere to the culture plate O/N and were subsequently repeatedly washed to remove non-adherent immune cells. Only after a long period of time (usually more than a month) we had enough cells that could be FACS sorted and would survive such procedure. These cultures seemed to favor the development of FRC-like cells. LEC development was extremely rare and BEC or DN stromal cell development was never observed. To fully disclose the origin of our stromal cell lines, we provide their phenotypic characterization in Figure 4—figure supplement 1.

5) A major point of this paper is that lymph node stromal cells support Treg maintenance however the data in Figure 4 indicate that the number of Tregs that persist in a culture with lymph node stromal cells is extremely low. How does this efficiency compare with dendritic cells? Representative flow cytometry data should accompany the graph in Figure 4.

The number of Tregs that persist in our cultures is low, indeed. Treg cell death in these cultures results from the absence of exogenously added survival factors, such as recombinant IL2. Recombinant IL2 was purposely left out as these experiments were designed to specifically address the role of self-antigen-presentation by stromal cells in the survival/maintenance of Tregs, which could have been masked by the addition of exogenous survival factors. Representative FACS plots of Treg recovery were added to Figure 4. We have not compared Treg persistence upon co-culture with stromal cells to co-culture with dendritic cells pulsed with OVA peptide. Given the critical role of TCR engagement in T cell survival, the higher levels of MHC-II surface expression on DCs and the need to add processed OVA peptides, such an experiment will likely result in higher T cell survival than when T cells are co-cultured with stromal cells expressing endogenously-derived OVA peptides. Indeed, even if OVA peptides were to be exogenously added to stromal cells and DCs, DCs would most likely outperform stromal cells, again given their higher MHC-II expression. Similar results are to be expected if full length OVA was to be added and antigen presentation would rely on antigen acquisition and processing, given the role of DCs as professional antigen-presenting cells.

6) Flow cytometry data for the efficiency of CD4 or CD8 T cell depletion in the donor tissues must be shown.

In addition to the graphs depicting CD4^+^ T cell depletion in the recipient’s own tissues, we now also provide data concerning CD4^+^ T cell depletion in the transplants in Figure 2—figure supplement 6. Determination of CD4^+^Foxp3^+^ T cell depletion in the transplants was not possible due to the very low number of CD4^+^ T cells retrieved from those lymph nodes. This precluded accurate analysis.

7) Statistical analysis of data in Figure 2—figure supplement 2 must be shown.

The statistical analysis of this data is now provided. In the revised manuscript this figure is Figure 2—figure supplement 5.

8) Figure 3, the y-axis is missing the values for cell counts.

In Figure 3 an offset histogram overlay is shown to more easily visualize the difference between Treg homeostatic proliferation in wild-type and MHC^-/-^ lymph node transplants. Such representation makes the axes of both curves distinct, thereby precluding the possibility to add a single scale on the Y-axis indicating cell counts. We have adjusted the label from ‘counts’ to ‘cell counts’, to be clearer. An indication of relative cell numbers can be deferred from Figure 3, showing that the frequency of Treg in MHC-II-/- lymph nodes is significantly reduced as compared to their frequency in wild-type transplants.

Reviewer 2 (pay particular attention to #1):

1) There is no characterization of cell population distributions and MHC-II expression patterns in transplanted WT LN or MHC-IIko LN. These are essential to understand the differences in T cell behavior in these two LN, and to support the assertions in the Results section and the significance of Figure 2—figure supplement 1.

We agree with this reviewer that analysis of MHC-II expression in transplanted lymph nodes is critical. We have included that analysis in Figure 2—figure supplement 1. Whereas dendritic cells and ETAC-like cells remain MHC-II positive, lymph node stromal cells exhibit a significant decrease in MHC-II expression in MHC-II-/- transplants as compared to wild-type transplants. The MHC-II+ stromal cells that we detected in MHC-II-/- transplants are likely to have acquired MHC-II expression from dendritic cells as shown by Dubrot et al, JEM 2014.

2) What is the basis for rejection of MHC-IIneg LN? The analyses of T cells in an MHC-IIneg LN are in an environment of inflammation and tissue damage, raising the question of whether their characteristics are causing the pathology or are affected by it, apart from any influence of Treg.

We have concluded that rejection of MHC-II-/- lymph node transplants was due to inefficient control of T cell activation caused by impaired maintenance of regulatory T cells. We acknowledge the fact that transplantation is accompanied by inflammation due to surgical trauma and tissue damage due to inefficient blood supply of the transplanted lymph node in the first days after transplantation. Such early inflammation and tissue damage can promote T cell activation. However, in MHC-II sufficient transplants such early T cell activation subsides as the healing process progresses. In MHC-II-/- transplants, the same T cell activation event seems to progress uncontrolled (due to Treg underrepresentation) and likely results in transplant rejection. In sum, we cannot exclude that T cell activation may contribute to and may get fuelled by tissue inflammation and damage. Indeed, the two events are interconnected and not easily separated.

3) The authors conclude that lack of MHC-II molecules on LNSC leads to diminished antigen-specific Treg maintenance/function and enables more robust activation of conventional CD4 and CD8 T cells. An alternative is based on the observation that MHC-II molecules are antigen-nonspecific ligands for the LAG-3 inhibitory receptor. This could affect the activation state of CD8 T cells and non-regulatory CD4 T cells independently of any influence on Treg representation. The data in right panels of Figure 2 and Figure 2—figure supplement 3 comparing WT and MHC-IIko LN in GK1.5 treated mice are strongly suggestive of this.

We agree with this reviewer that the MHC-II/LAG-3 axis can play an important role in our experimental system and thus contribute to the T cell activation seen in MHC-II-/- transplants independently of any Treg influence. Although we cannot rule out this possibility completely, our results regarding decreased Treg frequency (Figure 2—figure supplement 6) and inefficient Tregs expansion in MHC-II-/- lymph nodes (Figure 3) and improved survival of antigen-specific Tregs upon antigen recognition on stromal cells *in vitro* (Figure 4) and *in vivo* (Figure 5) strongly support a dominant role for Tregs in our assays. Indeed, in the in vitro assays we document an increase in antigen-specific Tregs induced by antigen-presentation by LECs that could be blocked by an anti-MHC-II antibody, and that non-antigen presenting FRCs were unable to induce.

Furthermore, when lack of MHC-II molecules would directly act on LAG-3 inhibitory receptors expressed on CD8 T cells and non-regulatory CD4 T cells, the enhanced activity state of CD8 cells, observed upon CD4 T cell depletion could not be easily explained. In our opinion, the data presented in Figure 2 and Figure 2—figure supplement 3 (now Figure 2—figure supplement 7) is more conclusive with CD4^+^ T cell depletion releasing a break, most likely imposed by Tregs, on CD8^+^ T cell activation.

4) Long-term LNSC lines are problematic, as the lineages have different survival capabilities and often lose differentiated characteristics. The authors acknowledge that their ova negative line is FRC like, while the ova pos line is LEC like. Thus, comparing their effects on T cells to establish a role for antigen (Figure 4) is very problematic. In Figure 4, even OVAneg LNSC lines are associated with a substantial representation of FoxP3+ cells in both bulk CD4 and OT-II T cells. Based on Figure 4—figure supplement 1, this is not due to proliferation, and is associated with upregulation of CD25 in CD8 T cells, but not OT-II cells. The authors state that this is an increase in survival of FoxP3+ cells, but FoxP3neg cells should also be examined, and there should be a zero time control to establish whether antigen is associated with selective survival or differentiation. Regardless, the distinct phenotypes of the cell lines will make it difficult to definitively conclude that antigen expression is responsible.

We agree with this reviewer that long-term maintenance of stromal cells is problematic because these cells lose many of the characteristics that they exhibit *in vivo*, namely chemokine production. However, the use of these cellular models as surrogate stromal cells is presently the best solution. In our hands, FACS-sorting stromal cells directly from lymph nodes always damaged them to the point that we could not culture them for the duration of our experiments.

As suggested by this reviewer, we added data characterizing the stromal cell lines used in our experiments in Figure 4—figure supplement 1 and data regarding the number of Foxp3^-^ and Foxp3^+^ CD4^+^ T cells in the input in Figure 4 and Figure 4—figure supplement 3. In Figure 4—figure supplement 3, we also added data regarding the behavior of CD4^+^Foxp3- conventional T cells, showing that these cells are not affected by co-culture with stromal cells in the same manner as CD4^+^Foxp3+ cells are. Importantly, the data regarding conventional T cells and Tregs were obtained in the exact same conditions, as we performed our stromal cell-T cell co-cultures with unfractionated CD4^+^ T cells containing both Foxp3^-^ and Foxp3^+^ cells.

Finally, in order to prove that antigen-presentation is the driving force that leads to increased numbers of Foxp3^+^ T cells in our assays, we have blocked antigen-presentation with the MHC-II specific antibody M5/114. As seen in Figure 4, blockade of MHC-II-mediated presentation prevented the K14mOVApos cell line to promote increased CD4^+^Foxp3+ OT-II T cell recovery to a point where “K14mOVApos cell line-induced” CD4^+^Foxp3+ OT-II T cell numbers were no longer distinct from “K14mOVApos cell line-induced” CD4^+^Foxp3+ wild-type T cell numbers. Thus, altogether, we believe that despite the intrinsic differences between our cell lines, i.e. between FRC- and LEC-like cell lines, our data accurately reflects the effect of stromal cell-mediated self-antigen presentation in Treg maintenance.

5) The authors interpret the results in Figure 5 as though only LNSC present antigen, but do not account for the possibility that ova positive LNSC transfer antigen to MHC-II positive hematopoetic cells in the same LN, or transfer preformed MHC-II antigen complexes. Thus, the results do not unambiguously establish that LNSC are responsible for any increase in representation of Treg or Treg subsets via direct presentation of antigen.

We agree with this reviewer that our *in vivo* experiments with K14mOVA lymph node transplants do not unambiguously establish that lymph node stromal cell-mediated antigen presentation is responsible for the increase in Treg representation and “immune tolerance”. However, they do establish that stromal cell-derived antigens can promote Treg expansion and thus provide protection. This can be caused by both direct and indirect (via other cells) antigen-presentation. Our *in vitro* experiments, however, clearly establish that lymph node stromal cells can present antigen in the context of MHC-I (as shown by others as well) and MHC-II molecules, with stromal cell MHC-II-mediated presentation contributing to the homeostasis of antigen-specific Tregs.

6) Figure 5 and Figure 5—figure supplement 1 show several instances in which changes in absolute numbers of Treg or Treg subsets change, but frequency or representation does not - or the converse. This must mean that other cell populations are also changing in response to characteristics of the transplanted LN. This needs to be clarified, as the manuscript text indicates that LN stroma contribute to the selective maintenance of antigen-specific.

There were instances in which the proportion of Foxp3+ OT-II T cells increased in K14mOVA lymph node transplant recipients while the absolute number of these cells did not. We attribute these discrepancies to the selective retention of OT-II T cells in K14mOVA transplants (see Figure 7 below – left panel). Antigen recognition in K14mOVA transplants possibly induces TCR-mediated stop signals (TCR-mediated CD69 upregulation would drive intracellular retention of the egress receptor S1P1 and thus trap OT-II T cells in the transplant (Shiow et al, Nature 2006)). Such an event induces a generalized increased in OT-II T cell numbers in K14mOVA transplants, including Foxp3^+^ OT-II Tregs (Figure 5), without necessarily changing the balance between Foxp3- and Foxp3+ OT-II T cells (Figure 5). In K14mOVA transplants, Foxp3+ OT-II T cells will receive enhanced survival factors, which will enhance their overall maintenance. As in the context of antigen-driven activation, CD4^+^Foxp3^+^ Tregs downregulate S1P1 at a slower pace as compared to conventional CD4^+^Foxp3^-^ T cells (Liu et al, NatImmunol 2009), they will egress from the K14mOVA transplants faster than their conventional counterparts and thus better populate distal lymph nodes. This leads to an increase in the percentage of CD4^+^Foxp3^+^ OT-II T cells in the endogenous lymph nodes of K14mOVA lymph node recipients. However, as the bulk of OT-II T cells are being actively sequestered in the K14mOVA transplants, and thus underrepresented in the endogenous lymph nodes of K14mOVA transplant recipients (see Figure 7 below – right panel), the increased frequency of CD4^+^Foxp3^+^ OT-II T cells in the endogenous lymph nodes of K14mOVA transplant recipients is not accompanied by an increase in the total number of these cells.

Furthermore, as far as our analysis of host-derived immune cells went, we have not observed significant differences between wild-type and K14mOVA transplant recipients.Author response image 1.

Reviewer 3 (comment on Figure 1 needs to be prioritized):

1) Figure 1: the authors claim a “selective absence of MHCII on LNSC from Tx MHCII-/- mice (transplanted in WT)”. Given the recent publication showing that MHCII are not only endogenously expressed by LNSC, but also acquired from DC, this result is quite surprising. It would be mandatory to show this result and compare with the situation in which DC do not express MHCII. For instance, show an absence of MHCII expression by LNSC from MHCII-/- LN Tx into chimeric mice (MHCII-/- BM into irradiated WT). Importantly, can the authors demonstrate any difference in Treg proliferation in the context of MHCII-/- LN Tx compared to endogenous distal LN?

We agree with this reviewer that analysis of MHC-II expression in transplanted lymph nodes is critical. We have now included such analysis in Figure 2—figure supplement 1. We show that whereas dendritic cells and ETAC-like cells remain MHC-II positive, lymph node stromal cells exhibit a significant decrease in MHC-II expression in MHC-II^-/-^ transplants as compared to wild-type transplants. As the reviewer points out, the MHC-II expression detected in MHC-II^-/-^ transplanted stromal cells is likely to have been acquired from dendritic cells (Dubrot et al. JEM 2014). The analysis suggested by the reviewer to address this point, unfortunately, could not be performed, as transfer of MHC-II^-/-^ bone marrow into wild-type recipients leads to severe disease and death of the chimeric animals (Marguerat et al, JImmunol 1999). Finally, as shown in the revised Figure 3—figure supplement 1, no difference in Treg proliferation in the endogenous lymph nodes of wild-type and MHC-II^-/-^ transplant recipients was observed. Thus, direct comparison of Treg proliferation between MHC-II^-/-^ transplants and the endogenous lymph nodes of MHC-II^-/-^transplant recipients, showed a significant reduction in Treg proliferation in the former (see Figure 8). Together, these data suggest that Treg proliferation was specifically impaired in MHC-II^-/-^ transplants.Author response image 2.

2) Figure 2 shows that T cell activation is increased in MHCII-/- LN Tx. As a demonstration of an effect happening locally in Tx LN, it is important to provide not only WT LN Tx, but distal endogenous LN fron MHCII-/- LN Tx mice as well.

We agree with this reviewer that additional clarification of the local effect of MHC-II deficiency is needed. We provided such data in Figure 2—figure supplement 4, which shows that no alterations in T cell activation were observed in the endogenous lymph nodes of MHC-II^-/-^ lymph node transplant recipients as compared to wild-type lymph node transplant recipients.

3) Figure 3 shows T cell (Treg and non Treg) proliferation 48h after OTII transfer. Is the observed effect still happening at later time points? Here as well, a distal endogenous LN should be provided. In addition, the division index ratio is quite confusing, since we do not have any indication of the proliferation rate of the cells. Please provide % of T cell (Treg and non Treg) proliferation, as well as % of Tregs. Moreover, dendritic cells were shown to be responsible for homeostatic Treg proliferation (Nussensweig's lab). How do the authors explain their results in that context? If, as shown, there are less Tregs in MHCII-/- LN Tx, we would have expected increased non Treg T cell proliferation, whereas opposite results are observed. Can the authors comment on that?

As suggested by this reviewer, in Figure 3—figure supplement 1 of the revised manuscript we provide data regarding the proliferation of conventional and Foxp3+ T cells separately, where one can see that Treg proliferation is significantly impaired in the absence of stromal cell-derived MHC-II, whereas proliferation of conventional T cells is not. Furthermore, we have also included data regarding T cell proliferation in the endogenous lymph nodes of transplant recipients, where one can observe that neither transplantation of wild-type nor MHC-II-/- lymph nodes interferes with the homeostatic proliferation of T cells in distant lymph nodes. In Figure 3, we also included the analysis of Treg frequency, which indicates that impaired proliferation of Tregs leads to a significant reduction of these cells in MHC-II^-/-^ transplants.

We have not analyzed DCs in the same experiments in which we determine Treg proliferation. However, we observed an increase in the frequency of DC in MHC-II^-/-^ transplants as compared to wild-type transplants (see Figure 9 below). This is in agreement with data by the Nussensweig's lab that showed that Treg depletion results in increased DC numbers by a Flt3 ligand-dependent mechanism (Lui et al, Science 2009) and likely reflects a compensatory mechanism in response to the diminished Treg numbers, which was however insufficient in the context of stromal MHC-II deficiency. In addition, one should mention that MHC-II expression in DCs is required to sustain Treg proliferation (Darasse-Jeze et al, JEM 2009) and in MHC-II^-/-^ lymph node transplants, infiltrating dendritic cells expressed normal levels of MHC-II molecules (Figure 2—figure supplement 1). Altogether, these data suggest that lymph node stromal cell MHC-II expression plays a major role in the homeostasis of Tregs, independently of dendritic cell MHC-II expression.Author response image 3.

Despite reduced Treg numbers, we did not observe an increase in CD4^+^ conventional T cell proliferation in the absence of stromal cell MHC-II expression (Figure 3—figure supplement 1). We agree that this may seem contradictory at first, given that Tregs will negatively regulate T cell expansion. However, it should be acknowledged that self-recognition is required for the maintenance of peripheral T cells as a whole (Takada and Jameson, NatRevImmunol 2009). Thus, lack of proper self-antigen presentation in the context of stromal cell MHC-II deficiency will interfere with antigen recognition by the overall CD4^+^ T compartment, thereby influencing the homeostatic reconstitution of the CD4+ conventional T cell pool as well. Notwithstanding, our data clearly shows that the homeostatic proliferation of CD4+ conventional T cell is less affected by the absence of stromal-derived MHC-II molecules than the homeostatic proliferation of CD4+Foxp3+ T cells.

4) Figure 4: The cell lines obtained and used by the authors (“FRC like OVAneg” and “LEC like OVApos”) represent a quite artificial system and are poorly relevant in the context of the paper (most experiments were performed in vivo). It is true that primary LEC / FRC cultures are contaminated by hematopoietic cells. However, in case the authors think these in vitro experiments are relevant for the paper, they need to sort LECs and FRC from those primary cultures to repeat the experiments. Furthermore, it is quite surprising that no CD8+ T cell proliferation was observed in these in vitro settings, since many studies have described early and transient CD8 T cell proliferation (followed by T cell deletion) after culture with by MHCI-Ag presenting LNSC.

We agree with this reviewer that stromal cell lines are a quite artificial system because long-term culture of lymph node stromal cells results in loss of many of their distinctive characteristics. Stromal cells sorted directly ex vivo from lymph nodes would be preferable, however, in our hands, stromal cell sorting results in highly damaged cells that invariably die in culture. The in vitro experiments are however of significance, as we show that in the absence of dendritic cells, antigen presentation by stromal cells clearly affects T cells in an antigen- and MHC-II-dependent manner. Regarding the lack of CD8^+^ T cell proliferation in our experiments, we can only speculate that model antigen expression driven by the human keratin14 promoter may be lower than when driven by the promoters used in other studies (iFABP-tOVA; GFAP-HA), thus resulting in less efficient peptide-MHC-I presentation.

5) Figures 5 and 6 are elegant and nice. The major issue I have is to understand whether the authors want to claim a local or a systemic effect of LNSC in dampening self-reactive T cell responses. Indeed, Figure 1 shows a local effect, whereas Figures 5 and 6 show a more general effect. Figure 5 shows a significant Treg increase in endogenous distal LN as well (although discrepancies exist between FACS dot plots (1.17% vs 1.7%) and histograms (about 3% vs >4% Treg). The authors need to clarify this point and, in case they think the effect is systemic, explain how it might work.

The effect of stromal cell MHC-II deficiency is local. It seems that the homeostatic maintenance of Tregs numbers (and possibly their suppressive function) requires continuous and appropriate recognition of peptide-MHC-II complexes in stromal cells. As Tregs egress from MHC-II^-/-^ transplants and recirculate through other secondary lymphoid organs, they will enter MHC-II sufficient environments that are able to rescue any imbalances to which Tregs may have been subjected in the transplant. Consequently, the Treg imbalance and T cell activation remains restricted to the transplant. Such local effect may, however, have systemic consequences when a given antigen is expressed only in one particular lymph node in which stromal cell-mediated presentation may become defective, resulting in the failure to generate antigen-specific Treg maintenance.

In contrast, the effect of K14mOVA lymph node transplantation is systemic. We attribute this to the recirculation of Tregs. In the K14mOVA transplant, lymph node stromal cell-derived OVA provides a survival advantage to OVA-specific Tregs allowing them to greatly expand. Such expansion coupled with Treg recirculation leads to a systemic overrepresentation of OVA-specific Tregs and thus to an increased ability to constrain immune responses.

The apparent discrepancy between the FACS plots and the summarizing graphs in Figure 5 arises from the selection of the two FACS plots with lower frequency of Foxp3^+^CD62L^+^ Tregs. The range of Foxp3^+^CD62L^+^ Tregs frequencies in wild-type transplants and K14mOVA transplants is 1.17-3.74% and 1.70-7.31%, respectively. To more faithfully represent our data, in the revised Figure, we have replaced those plots by plots showing percentages of Foxp3^+^CD62L^+^ Tregs closer to the averages observed.